# JunB is essential for IL-23-dependent pathogenicity of Th17 cells

Zafrul Hasan[1,*], Shin-ichi Koizumi[1,*], Daiki Sasaki[1,*], Hayato Yamada[1], Nana Arakaki[2], Yoshitaka Fujihara[3], Shiho Okitsu[1], Hiroki Shirahata[1] & Hiroki Ishikawa[1]

CD4$^+$ T-helper cells producing interleukin-17 (IL-17), known as T-helper 17 (T$_H$17) cells, comprise heterogeneous subsets that exhibit distinct pathogenicity. Although pathogenic and non-pathogenic T$_H$17 subsets share a common RORγt-dependent T$_H$17 transcriptional programme, transcriptional regulatory mechanisms specific to each of these subsets are mostly unknown. Here we show that the AP-1 transcription factor JunB is critical for T$_H$17 pathogenicity. JunB, which is induced by IL-6, is essential for expression of RORγt and IL-23 receptor by facilitating DNA binding of BATF at the *Rorc* locus in IL-23-dependent pathogenic T$_H$17 cells, but not in TGF-β1-dependent non-pathogenic T$_H$17 cells. *Junb*-deficient T cells fail to induce T$_H$17-mediated autoimmune encephalomyelitis and colitis. However, JunB deficiency does not affect the abundance of gut-resident non-pathogenic T$_H$17 cells. The selective requirement of JunB for IL-23-dependent T$_H$17 pathogenicity suggests that the JunB-dependent pathway may be a therapeutic target for autoimmune diseases.

[1] Immune Signal Unit, Okinawa Institute of Science and Technology Graduate University, 1919-1 Tancha, Onna-son, Okinawa 904-0495, Japan. [2] DNA Sequencing Section, Okinawa Institute of Science and Technology Graduate University, 1919-1 Tancha, Onna-son, Okinawa 904-0495, Japan. [3] Research Institute for Microbial Diseases, Osaka University, 3-1 Yamadaoka, Suita, Osaka 565-0871, Japan. * These authors contributed equally to this work. Correspondence and requests for materials should be addressed to H.I. (email: hiroki.ishikawa@oist.jp).

nterleukin-17 (IL-17)-producing T-helper 17 (T$_H$17) cells, exhibiting heterogeneous pathogenicity, serve diverse biological functions[1–3]. Pathogenic T$_H$17 cells have a central function in autoimmune and chronic inflammatory diseases[4–6], whereas non-pathogenic T$_H$17 cells, which accumulate in the gut at steady state, are probably involved in gut homoeostasis and host defence[7–9]. Pathogenicity of T$_H$17 cells is associated with cytokine signals. A subset of T$_H$17 cells induced in the presence of transforming growth factor β1 (TGF-β1) and IL-6 (hereafter referred to as T$_H$17(β))[10–12] is non-pathogenic, because transfer of the cells into mice induces weak or no experimental autoimmune encephalomyelitis (EAE)[13,14]. On the other hand, another subset of T$_H$17 cells generated in the presence of IL-6, IL-1β and IL-23 (hereafter referred to as T$_H$17(23)) is highly pathogenic, and transfer of these cells into mice induces severe EAE[13,14]. Furthermore, it is broadly accepted that IL-23 is needed for T$_H$17 cell pathogenicity and that it can induce pathogenicity in previously non-pathogenic T$_H$17 cells[15–18]. Indeed, mice deficient in IL-23 signalling are resistant to EAE and chronic colitis[15,17,19,20], but they have normal numbers of gut-resident T$_H$17 cells at steady state[21]. The critical function of IL-23 signalling for T$_H$17 pathogenicity is also evidenced by a correlation between IL-23 mutation and human autoimmune diseases[22].

Pathogenic and non-pathogenic T$_H$17 cells differentially express a number of genes. For example, anti-inflammatory IL-10 is specifically induced in T$_H$17(β) cells, whereas inflammatory granulocyte–macrophage colony-stimulating factor (GM-CSF) is preferentially produced by T$_H$17(23) cells[13,14,23–25] to contribute to the pathogenicity of T$_H$17 cells[26,27]. However, pathogenic and non-pathogenic T$_H$17 cells express a subset of molecules (T$_H$17 signature genes) comparably, including *Il17a, Il17f* and *Il23 receptor (Il23r)*[13,23,25]. Induction of these T$_H$17 signature genes is regulated by transcription factors including basic leucine zipper ATF-like transcription factor (BATF), interferon regulatory factor 4 (IRF4), signal transducer and activator of transcription 3 (STAT3), and RORγt in both pathogenic and non-pathogenic T$_H$17 cells[28–32]. BATF and IRF4 are induced by T-cell receptor signalling and bind to loci of a large number of genes, including T$_H$17 signature genes, where they promote chromatin accessibility[29,33,34]. Under T$_H$17-polarizing conditions, cytokine signals activate STAT3 and induce RORγt, both of which bind to loci of T$_H$17 genes occupied by BATF and IRF4 and activate T$_H$17 gene transcription[28,29]. IL-6 activates STAT3 through JAK-mediated phosphorylation[35,36]. However, it is unclear how RORγt induction is regulated by different combinations of cytokines under T$_H$17(β)- and T$_H$17(23)-polarizing conditions.

Here we show that, in an RNAi screen of transcription factors involved in IL-23 signalling, an AP-1 transcription factor, JunB, is required for IL-23-dependent gene induction. Additional analyses show that JunB is required for induction of RORγt in pathogenic T$_H$17(23) cells, but not in non-pathogenic T$_H$17(β) cells. Mechanistically, JunB facilitates DNA binding of BATF, IRF4 and STAT3 at multiple gene loci including *Rorc* (encoding RORγt) and *Il17a* under T$_H$17(23)-polarizing conditions. Furthermore, we show that JunB is essential for pathogenicity of T$_H$17 cells in EAE and colitis models, but it is not required for generation of non-pathogenic, gut-resident T$_H$17 cells. These data suggest that the JunB-dependent pathway is required for IL-23-dependent pathogenicity of T$_H$17 cells.

## Results

**Identification of JunB as a regulator of IL-23 signalling.** TGF-β1 signalling is associated with non-pathogenic T$_H$17

differentiation, whereas IL-23 signalling facilitates pathogenicity of T$_H$17 cells[13,15–17,19]. However, transcriptional mechanisms underlying control of T$_H$17 pathogenicity in the presence of these cytokines remain to be fully determined. To better understand IL-23-dependent transcriptional regulation, we attempted to identify transcription factors responsible for expression of genes promoted by IL-23 signalling in T$_H$17 cells. Based on published microarray data[13,23], we selected 263 transcription factors that are highly expressed in T$_H$17 cells (Supplementary Data 1). Retroviruses expressing shRNAs against these transcription factors were individually transduced into T$_H$17 cells generated under pathogenic T$_H$17(23)-polarizing conditions (in the presence of IL-6, IL-1β and IL-23). To evaluate the effect of transcription factor knockdown on IL-23-dependent signalling, we measured levels of *ectonucleotide pyrophosphate/phosphodiesterase 2 (Enpp2)* mRNA because we found that induction of *Enpp2* was significantly facilitated by IL-23 stimulation (Supplementary Fig. 1a). *Enpp2* induction was most heavily diminished by knockdown of an AP-1 transcription factor, JunB (Supplementary Fig. 1b). RNAi ablation of JunB also significantly reduced expression of a T$_H$17 signature molecule, *Il23r* (Supplementary Fig. 1c). JunB interacts with another AP-1 family member, BATF, an essential transcription factor for T$_H$17 differentiation[31,33,34], suggesting that JunB might be involved in T$_H$17 differentiation; however, the physiological functions of JunB in T$_H$17 differentiation remain unknown.

**JunB is induced in T$_H$17 cells.** We first examined JunB expression in T$_H$17 cells differentiated *in vitro*. Immunoblot analysis of JunB showed that naive CD4$^+$ T cells activated under T$_H$17(β)-polarizing conditions (in the presence of TGF-β1 and IL-6) or T$_H$17(23)-polarizing conditions expressed higher levels of JunB compared to T cells activated under neutral conditions (in the absence of cytokines) (T$_H$0) or induced T regulatory (iTreg)-polarizing conditions (in the presence of TGF-β1 and IL-2) (Fig. 1a). There was no detectable JunB expression in naive CD4$^+$ T cells (Supplementary Fig. 1d). Quantitative reverse transcription–polymerase chain reaction (qRT–PCR) analysis also showed an increase in *Junb* mRNA levels in both T$_H$17(β) and T$_H$17(23) cells (Fig. 1b). We also found that IL-6 stimulation was sufficient to augment JunB expression, whereas neither IL-1β nor IL-23 signalling significantly affected JunB expression in activated CD4$^+$ T cells (Fig. 1c,d). IL-6 signalling is mediated by STAT3 (refs 35,36). Indeed, JunB induction was severely impaired in *Stat3*-deficient cells under T$_H$17(β)-polarizing conditions (Fig. 1e), indicating that STAT3 is required for JunB induction in T$_H$17 cells. These results suggest that JunB expression is facilitated in T$_H$17 cells in an IL-6- and STAT3-dependent manner.

**JunB regulates T$_H$17 differentiation *in vitro*.** To determine the importance of JunB in T$_H$17 differentiation, we generated T-cell-specific *Junb*-deficient (Cd4$^{Cre}$Junb$^{fl/fl}$) mice (Supplementary Fig. 2). Loss of JunB did not affect the abundance of CD4$^+$ and CD8$^+$ T cells, nor did it affect that of naive (CD62L$^{hi}$CD44$^{lo}$), effector (CD62L$^{hi}$CD44$^{hi}$) and memory (CD62L$^{lo}$CD44$^{hi}$) CD4$^+$ T-cell populations in lymph nodes (LNs) and spleens (Supplementary Fig. 3).

To evaluate the function of JunB in T$_H$17 differentiation *in vitro*, we activated *Junb*-deficient naive CD4$^+$ T cells under T$_H$17-polarizing conditions. Notably, IL-17A expression was severely diminished in *Junb*-deficient CD4$^+$ T cells under pathogenic T$_H$17(23)-polarizing conditions (Fig. 2a). However, a substantial number of *Junb*-deficient CD4$^+$ T cells produced IL-17A, albeit less than that produced by control cells, under

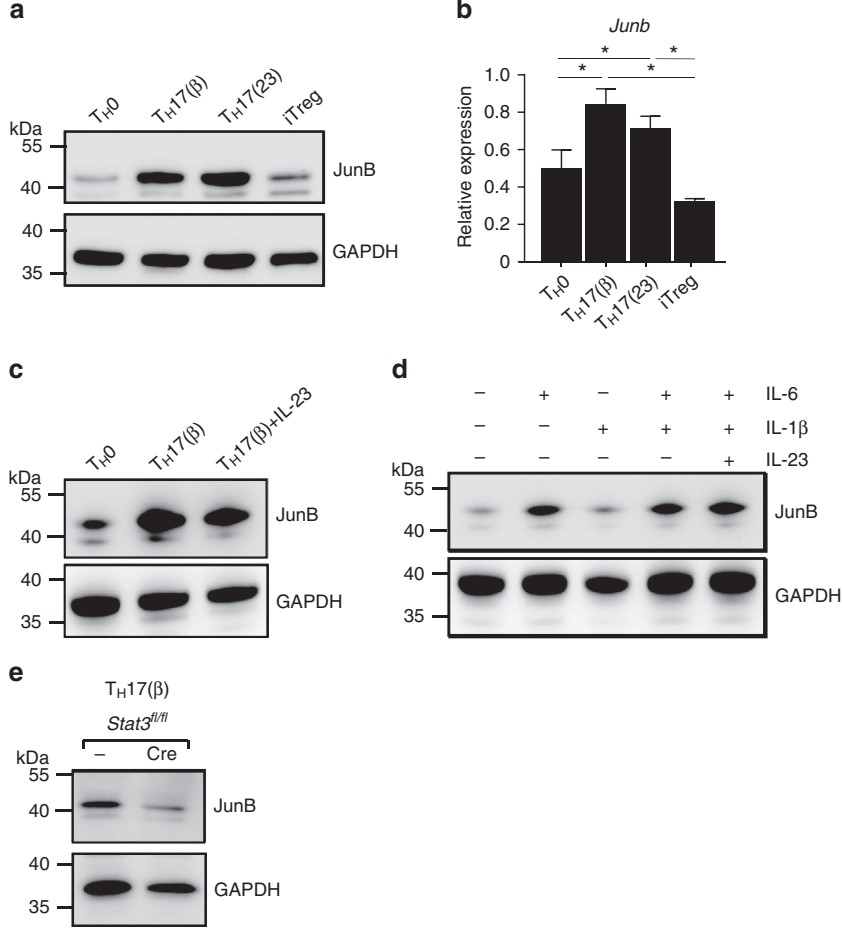

**Figure 1 | JunB is induced in T$_H$17 cells.** (**a,b**) Naive CD4$^+$ T cells were activated with anti-CD3 and anti-CD28 antibodies in the presence of TGF-β1 and IL-6 (T$_H$17(β)), IL-6, IL-1β and IL-23 (T$_H$17(23)), TGF-β1 and IL-2 (iTreg) or in the absence of cytokines (T$_H$0) for 60 h. JunB protein (**a**) and mRNA (**b**) were detected by immunoblot and qRT–PCR, respectively. mRNA results were normalized to *Hprt* mRNA. Error bars indicate s.d. (*n* = 4). Asterisks indicate significant differences (*P* < 0.05) by unpaired two-tailed Student's *t*-test. (**c,d**) Immunoblot analysis of JunB in CD4$^+$ T cells activated in the presence of the indicated cytokines for 60 h. (**e**) Naive CD4$^+$ T cells from *Stat3$^{fl/fl}$* mice were activated, infected with Cre-expressing retrovirus, then cultured under T$_H$17(β)-polarizing conditions for 60 h. JunB was detected by immunoblot analysis. (**a,c,e**) Data represent two independent experiments. (**b,d**) Data represent three independent experiments.

non-pathogenic T$_H$17(β)-polarizing conditions (Fig. 2a). Loss of JunB promoted production of a T$_H$1 signature cytokine, IFN-γ, under T$_H$17(23) conditions (Fig. 2a). Moreover, expression of a T$_H$17 pathogenic cytokine, GM-CSF, was also augmented in the absence of JunB (Supplementary Fig. 4a). Consistent with its impact on production of IL-17A, induction of an essential T$_H$17 transcription factor, RORγt, was severely impaired in *Junb*-deficient T cells under T$_H$17(23)-polarizing conditions, but less strikingly under T$_H$17(β)-polarizing conditions (Fig. 2b and Supplementary Fig. 4b). Normal mRNA expression of IL-6 receptor was observed in *Junb*-deficient cells under T$_H$17(β) conditions, while it was slightly elevated under T$_H$17(23) conditions (Supplementary Fig. 4c), suggesting that the defective T$_H$17(23) differentiation in *Junb*-deficient cells may not be due to impaired IL-6 receptor expression.

Remarkably, expression of the Treg master transcription factor, forkhead box protein 3 (Foxp3)[37], significantly increased in *Junb*-deficient CD4$^+$ T cells under T$_H$17(β) conditions, but not under T$_H$17(23) or iTreg conditions (Supplementary Fig. 4b,d), suggesting that JunB is involved in IL-6-mediated suppression of TGF-β-dependent Foxp3 induction. In addition, JunB deficiency also resulted in induction of Foxp3 and a slight reduction of IL-17A production under other T$_H$17-polarizing conditions (the

presence of IL-6 and TGF-β3; T$_H$17(β3)) (Fig. 2c and Supplementary Fig. 4e). Foxp3 suppresses IL-17A production by antagonizing functions of RORγt[38,39]. Indeed, the abundance of IL-17A-expressing cells in a population with low Foxp3 expression (Foxp3-low population) was greater than in a population with high Foxp3 expression (Foxp3-high population) in *Junb*-deficient cells activated under T$_H$17(β) conditions (Supplementary Fig. 4f), suggesting that defective IL-17A production in *Junb*-deficient T$_H$17(β) cells might be partly due to aberrant induction of Foxp3.

Since IL-23 promotes pathogenicity of T$_H$17(β) cells[14], we investigated the function of JunB in IL-23-stimulated T$_H$17(β) cells. When we activated CD4$^+$ T cells in the presence of TGF-β1 and IL-6, with or without IL-23, IL-23 did not affect IL-17A and IFN-γ production in either *Junb*-deficient or control cells (Supplementary Fig. 4g). We next examined the impact of JunB deficiency on IL-23-stimulated T$_H$17(β) cells in the absence of TGF-β1. We activated *Junb*-deficient T cells under T$_H$17(β) conditions for 3 days and sorted them into IL-17-high or IL-17-low populations using an IL-17-capture method for further culturing with IL-23 alone. Although JunB deficiency did not affect expression of IL-17A on re-stimulation immediately after sorting, further culturing of IL-17-high cells with IL-23

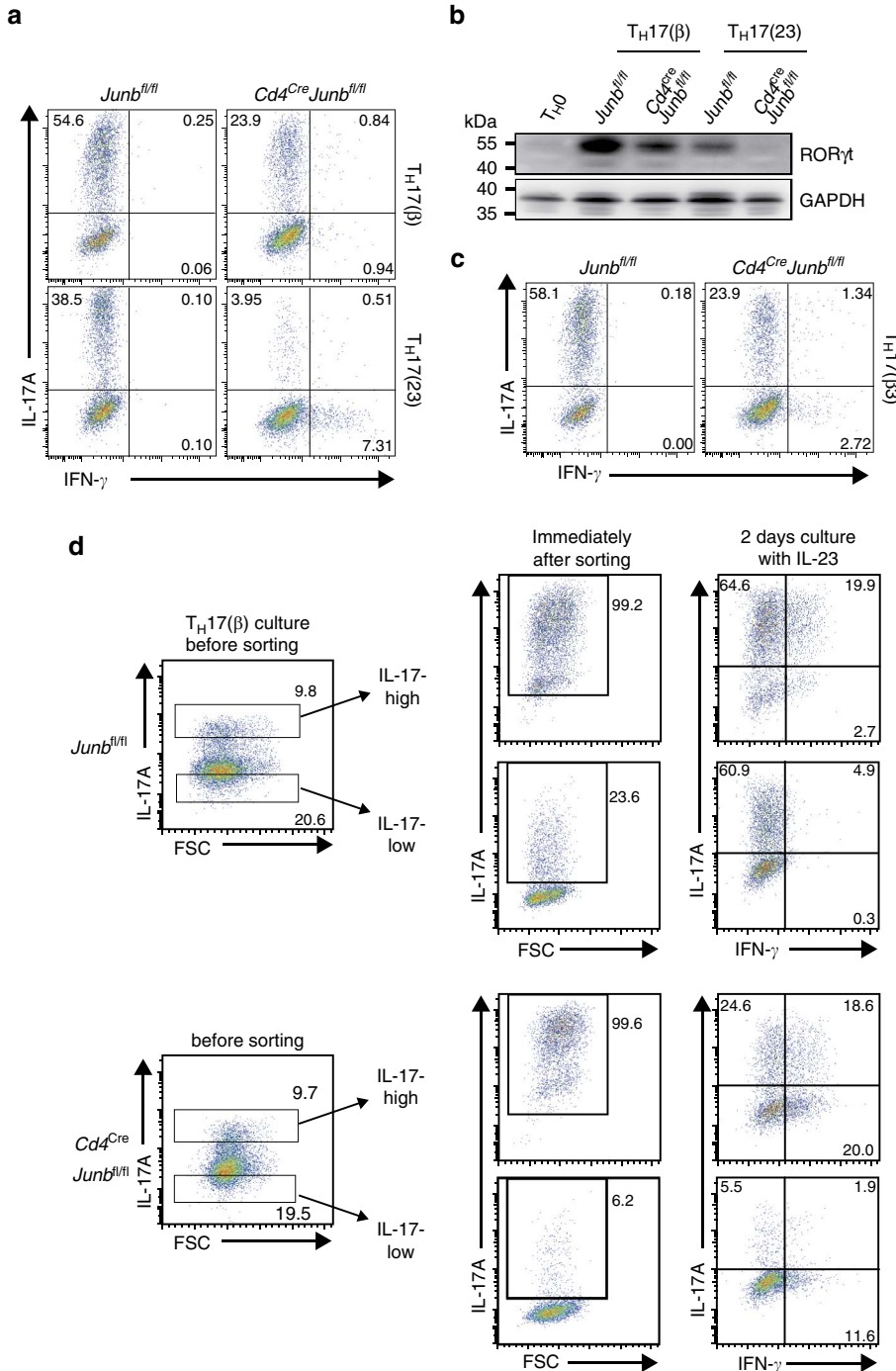

**Figure 2 | JunB regulates T_H17 differentiation.** (**a,b**) Naive CD4$^+$ T cells from *Cd4$^{Cre}$Junb$^{fl/fl}$* and control *Junb$^{fl/fl}$* mice were activated under T_H17(β)- or T_H17(23)-polarizing conditions for 3 days, and expression of IL-17A and IFN-γ was analysed by flow cytometry (**a**). RORγt was detected by immunoblot analysis (**b**). (**c**) Naive CD4$^+$ T cells from *Cd4$^{Cre}$Junb$^{fl/fl}$* and control *Junb$^{fl/fl}$* mice were activated in the presence of TGF-β3 and IL-6 (T_H17(β3)) for 3 days, and expression of IL-17A and IFN-γ was analysed by flow cytometry. (**d**) *Cd4$^{Cre}$Junb$^{fl/fl}$* and control cells were activated under T_H17(β)-polarizing conditions for 3 days, and IL-17-high and IL-17-low populations were sorted using the IL-17-capture method. Enrichment of IL-17-high cells was confirmed by detecting IL-17A expression on re-stimulation immediately after sorting. Cells were then cultured in the presence of IL-23 alone for another 2 days. Production of IL-17A and IFN-γ was assessed by flow cytometry. (**a,c**) Data represent three independent experiments. (**b,d**) Data represent two independent experiments.

significantly decreased the abundance of IL-17A-expressing cells in *Junb*-deficient cells, but not in controls (Fig. 2d). The data also showed a marked increase in the proportion of IFN-γ single-producing cells in the absence of JunB (Fig. 2d). Interestingly, IL-23 facilitated IL-17A expression in the IL-17-low populations of control cells, but not in *Junb*-deficient cells (Fig. 2d). Thus, JunB

seems to have an important function in IL-23-dependent maintenance of T_H17 cells in the absence of TGF-β1.

**JunB-dependent transcriptional regulation in T_H17 cells.** To further clarify functions of JunB in T_H17 differentiation, we performed a microarray analysis of *Junb*-deficient

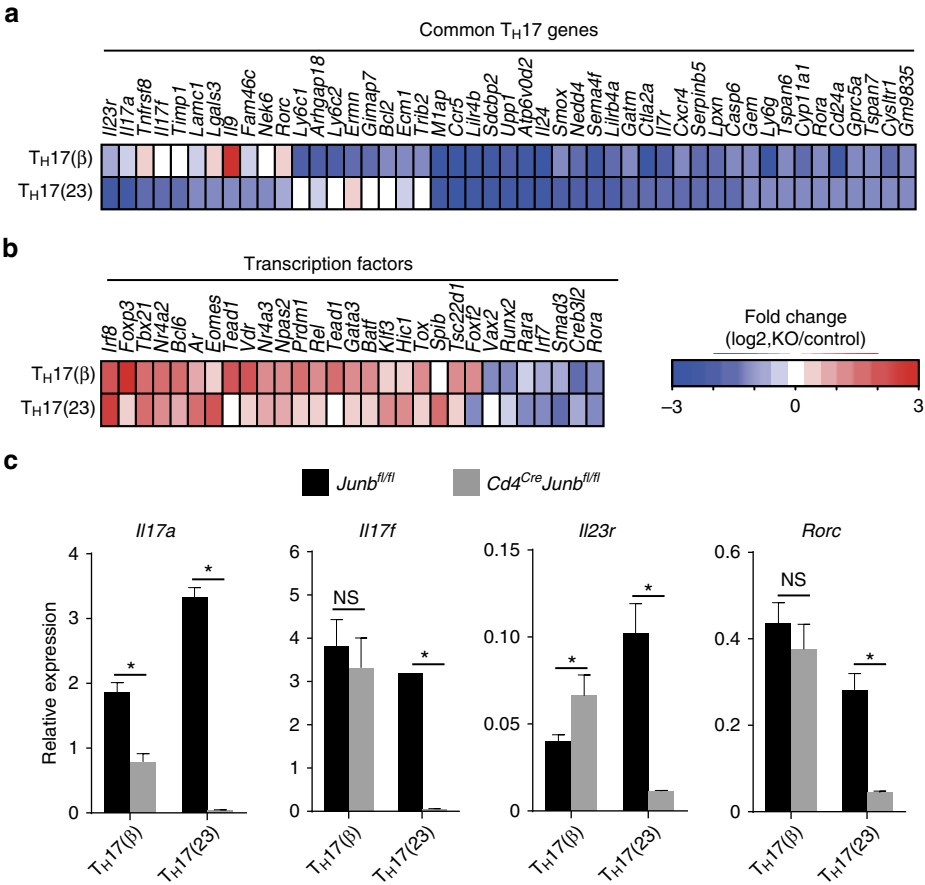

**Figure 3 | JunB-dependent transcriptional regulation in T$_H$17 cells.** (**a**,**b**) Microarray analysis of $Cd4^{Cre}Junb^{fl/fl}$ and control cells activated under T$_H$0-, T$_H$17(β)- or T$_H$17(23)-polarizing conditions for 60 h. Heat map data show fold changes of expression in $Cd4^{Cre}Junb^{fl/fl}$ (KO) versus ($Junb^{fl/fl}$ (control)) cells for common T$_H$17 genes (**a**), and genes categorized as transcription factors (**b**). Only genes that showed a significant change in $Cd4^{Cre}Junb^{fl/fl}$ cells compared to controls are shown. Common T$_H$17 genes represent genes significantly upregulated in both T$_H$17(β) and T$_H$17(23) compared to T$_H$0 (as in Supplementary Fig. 5a). (**c**) $Cd4^{Cre}Junb^{fl/fl}$ and control cells were activated under T$_H$17(β)- or T$_H$17(23)-polarizing conditions for 84 h, and mRNA expression of $Il17a$, $Il17f$, $Il23r$ and $Rorc$ was analysed by qRT–PCR. Results were normalized to $Hprt$ mRNA. Error bars indicate s.d. Asterisks indicate significant differences ($P < 0.05$) by unpaired two-tailed Student's $t$-test. NS, not significant. (**a**,**b**) Data represent two independent experiments. (**c**) Data represent three independent experiments.

and control CD4$^+$ T cells activated under T$_H$17(β)- or T$_H$17(23)-polarizing conditions. Our data showed that expression of only a small subset (11 out of 188 genes) of common T$_H$17 genes, induced in both T$_H$17(β) and T$_H$17(23) cells, such as $Il17a$, $Il17f$ and $Rorc$, was significantly reduced by the loss of JunB, specifically under T$_H$17(23)-polarizing conditions (Fig. 3a and Supplementary Fig. 5a). Moreover, JunB deficiency also reduced expression of 28 of the common T$_H$17 genes, such as $Rora$, $Il24$ and $Ccr5$, under both T$_H$17(β) and T$_H$17(23) conditions (Fig. 3a). In addition to aberrant induction of $Foxp3$ in T$_H$17(β) cells, JunB deficiency also resulted in dysregulation of expression of the T$_H$1 master transcription factor, $Tbx21$ (Fig. 3b), which is consistent with the reported JunB-dependent suppression of $Tbx21$ induction in T$_H$2 cells[40,41]. qRT–PCR data confirmed the defective induction of T$_H$17 signature genes $Il17a$, $Il17f$, $Il23r$ and $Rorc$ mRNAs in $Junb$-deficient CD4$^+$ T cells under T$_H$17(23)-polarizing conditions (Fig. 3c). On the other hand, JunB deficiency resulted in a significant increase in $Foxp3$, but it had only minor effects, if any, on expression of $Il17a$, $Il17f$, $Rorc$ and $Il23r$ under T$_H$17(β) or T$_H$17(β3) conditions (Fig. 3c and Supplementary Fig. 5b,c).

We further investigated effects of JunB deficiency on gene expression profiles in IL-17A-high and IL-17-low populations induced under T$_H$17(β) conditions. We found that JunB deficiency decreased expression of a subset of common T$_H$17 genes, such as $Ccr5$ in both IL-17-high and IL-17-low populations in a similar manner (Supplementary Fig. 5d,e). However, expression of $Foxp3$ was significantly higher in the IL-17-low population than the IL-17-high population (Supplementary Fig. 5e), suggesting that Foxp3 may suppress IL-17 expression at the transcriptional level.

To examine whether the impairment of RORγt induction is due to aberrant induction of T-bet in $Junb$-deficient cells under T$_H$17(23) conditions, we analysed RORγt and T-bet expression at the single-cell level by flow cytometry. The data showed that RORγt expression was significantly reduced not only in the T-bet-high population, but also in the T-bet-low population in $Junb$-deficient cells under T$_H$17(23) conditions (Supplementary Fig. 5f), suggesting that JunB regulates expression of RORγt and T-bet independently. Collectively, these results suggest that JunB regulates a limited subset of T$_H$17 genes in a context-dependent manner, and that activation of the RORγt-dependent core T$_H$17 transcriptional programme relies on JunB in IL-23-dependent pathogenic T$_H$17 cells, but not in TGF-β1-dependent non-pathogenic T$_H$17 cells.

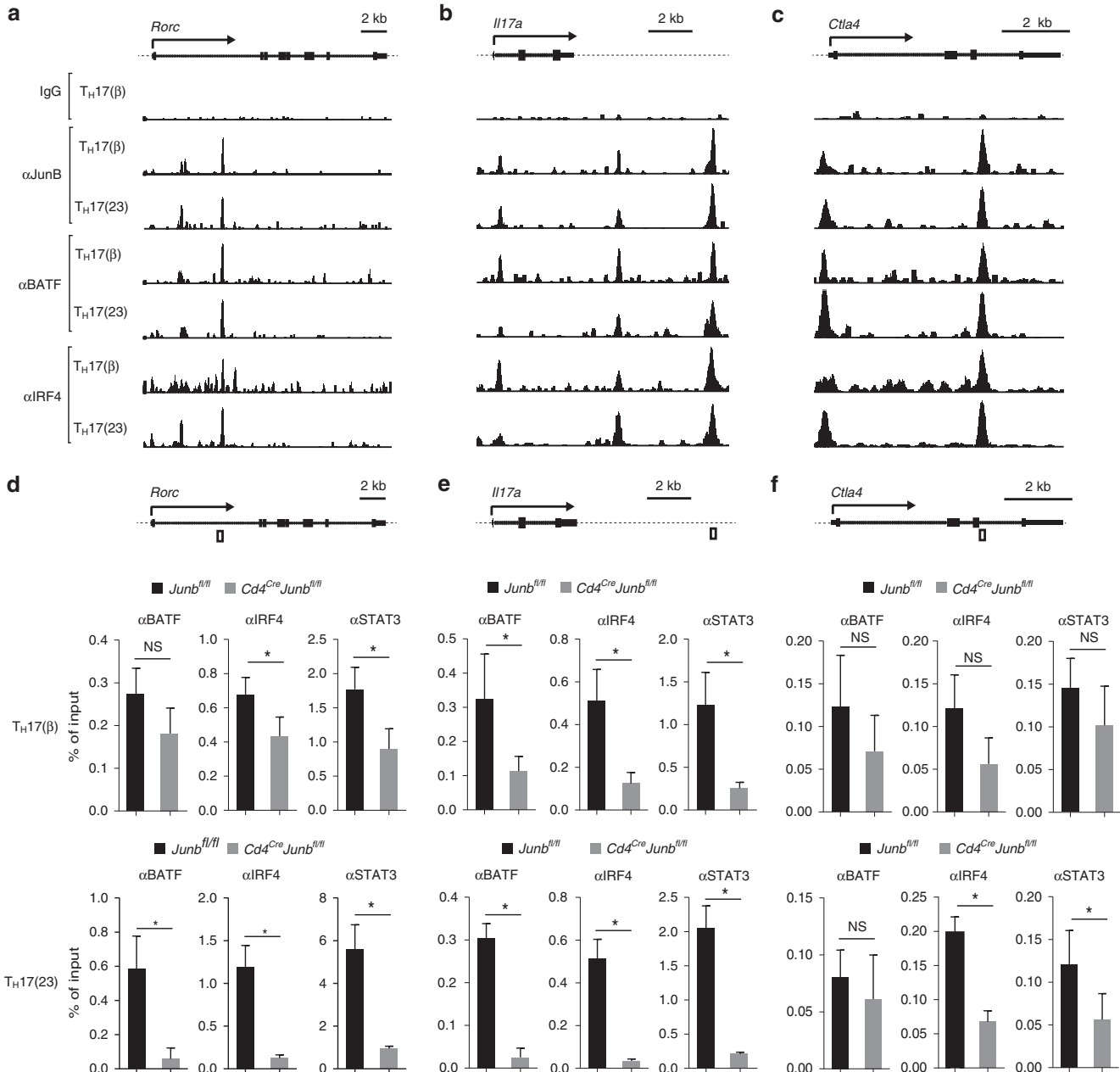

**Figure 4 | JunB promotes DNA binding of BATF and IRF4.** (**a**–**c**) Naive CD4+ T cells from wild-type mice were activated under T$_H$17(β)- or T$_H$17(23)-polarizing conditions for 60 h and subjected to ChIP-seq analysis using JunB, BATF and IRF4 antibodies. *Rorc* (**a**), *Il17a* (**b**) and *Ctla4* (**c**) loci are shown. Schematic representations at the tops of panels indicate transcription start sites (arrows), exons (filled boxes), introns (solid lines) and non-coding regions (dashed lines). (**d**–**f**) Naive CD4+ T cells from *Cd4$^{Cre}$Junb$^{fl/fl}$* or control mice were activated under T$_H$17(23)-polarizing conditions for 84 h and subjected to ChIP analysis using BATF, IRF4 and STAT3 antibodies. Eluted DNA was analysed by qPCR using primers to detect gene regions of *Rorc* (**d**), *Il17a* (**e**) and *Ctla4* (**f**). Schematic representations show regions detected by PCR (open boxes). Error bars indicate s.d. Asterisks indicate significant differences (*P* < 0.05) by unpaired two-tailed Student's *t*-test. NS, not significant. Data represent two independent experiments.

**JunB regulates DNA binding of BATF and IRF4.** To further insight into JunB-dependent transcriptional regulation in T$_H$17 cells, we investigated genome-wide JunB-DNA binding in T$_H$17 cells using chromatin immunoprecipitation sequencing (ChIP-seq) analysis with anti-JunB antibody. Consistent with the reported interaction between JunB and BATF[31,33,34], we found that JunB co-localized with BATF and IRF4 at loci of not only T$_H$17 signature genes, including *Rorc*, *Il17a* and *Il23r*, but also *Tbx21*, under both T$_H$17(β) and T$_H$17(23) conditions in a similar manner (Fig. 4a,b and Supplementary Fig. 6a,b), suggesting that JunB may directly regulate transcription of these genes. JunB,

BATF and IRF4 were also enriched at loci of genes, induction of which was independent of JunB, such as *Ctla4* (Fig. 4c and Supplementary Fig. 5b). A non-pathogenic signature cytokine gene, *Il10*, expression of which is regulated by BATF and IRF4 (ref. 33), was also induced in a JunB-independent manner in T$_H$17(β) cells (Supplementary Fig. 5b). However, at the *Il10* locus, there was significant enrichment of BATF, IRF4 and JunB in both T$_H$17(β) and T$_H$17(23) cells (Supplementary Fig. 6c), which implies that JunB co-localizes with BATF and IRF4 at a large number of gene loci, but that a limited subset of the genes is regulated by JunB.

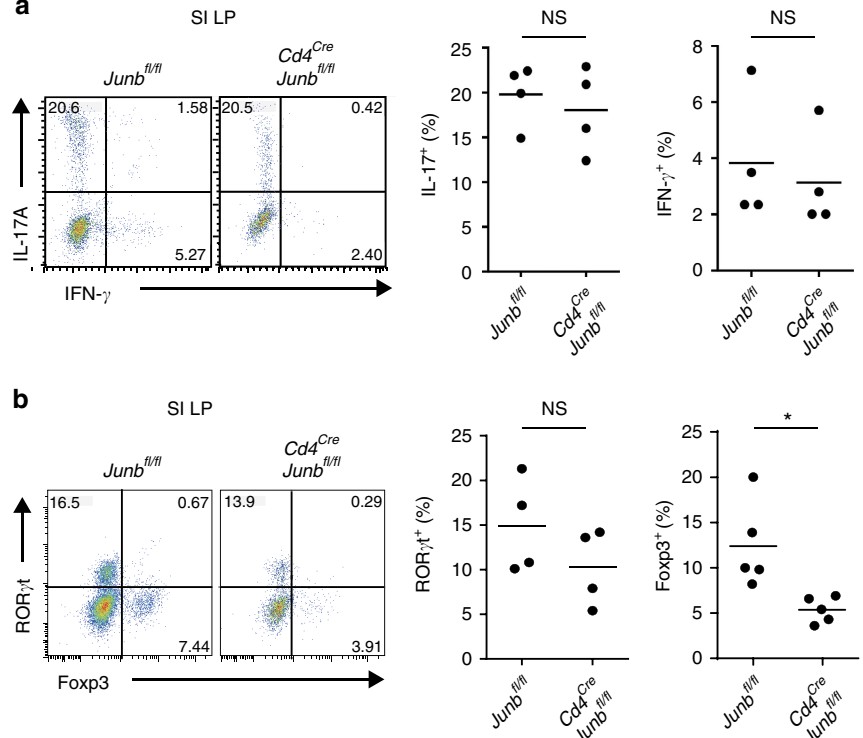

**Figure 5 | JunB-independent production of gut-resident $T_H17$ cells.** (**a**,**b**) Frequencies of CD4$^+$ cells expressing IL-17A and IFN-γ (**a**) or RORγt and Foxp3 (**b**) were analysed by flow cytometry ($n = 4$). Horizontal bars indicate the means. Asterisks indicate significant differences ($P < 0.05$) by unpaired two-tailed Student's $t$-test. SI LP, small intestine lamina propria. NS, not significant. Data represent two independent experiments.

Blimp1 promotes pathogenic $T_H17$ generation, whereas CD5L is associated with non-pathogenic $T_H17$ differentiation[42,43]. Although we could not detect induction of these molecules under our $T_H17$-polarizing conditions, we found that JunB, together with BATF and IRF4, was enriched at the *Prdm1* (encoding Blimp1) locus in both $T_H17(β)$ and $T_H17(23)$ cells (Supplementary Fig. 6d). Furthermore, DNA binding of JunB, BATF and IRF4 at the *Cd5l* locus was observed specifically in $T_H17(23)$ cells (Supplementary Fig. 6e). These findings imply that JunB might also be involved in control of these regulators of $T_H17$ pathogenicity, probably at later stages of $T_H17$ differentiation.

We next evaluated the impact of JunB on DNA binding of BATF, IRF4 and STAT3 during $T_H17$ differentiation, using ChIP–PCR analysis. Loss of JunB considerably diminished DNA binding of BATF at the *Rorc* locus, under $T_H17(23)$ conditions, but not under $T_H17(β)$ conditions (Fig. 4d). JunB deficiency also resulted in a great reduction of binding of IRF4 and STAT3 at the *Rorc* locus under $T_H17(23)$ conditions, but the effect was relatively small under $T_H17(β)$ conditions (Fig. 4d). JunB deficiency also impaired binding of BATF, IRF4 and STAT3 to the *Il17a* locus under both $T_H17(β)$ and $T_H17(23)$ conditions (Fig. 4e), which is consistent with the positive role of JunB in IL-17A production, even under $T_H17(β)$ conditions. However, JunB deficiency did not affect the DNA binding of BATF, even though it only slightly decreased that of IRF4 and STAT3, at the *Ctla4* locus under $T_H17(23)$-polarizing conditions (Fig. 4f). Collectively, these data suggest that JunB may be critical for DNA binding of BATF, IRF4 and STAT3 at the *Rorc* locus in IL-23-dependent $T_H17$ cells, but not in TGF-β1-dependent non-pathogenic $T_H17$ cells.

**JunB-independent generation of gut-resident $T_H17$ cells.** We next evaluated the importance of JunB for *in vivo* $T_H17$ differentiation. We found that a substantial proportion (10–20%) of CD4$^+$ T cells expressed IL-17A in the small intestinal lamina propria (SI LP) in *Cd4$^{Cre}$Junb$^{fl/fl}$* mice, comparable to control (*Junb$^{fl/fl}$*) mice (Fig. 5a and Supplementary Fig. 7a). Furthermore, a proportion of RORγt-expressing CD4$^+$ T cells was similar in *Cd4$^{Cre}$Junb$^{fl/fl}$* and control mice (Fig. 5b), suggesting that SI LP $T_H17$ cells are more likely to be generated in a JunB-independent manner. Contrary to our *in vitro* data, showing upregulation of Foxp3 in *Junb*-deficient $T_H17(β)$ cells, loss of JunB did not affect the abundance of CD4$^+$Foxp3$^+$RORγt$^+$ cells in SI LP (Fig. 5b), suggesting that Foxp3 induction may be suppressed in a JunB-independent manner or that Foxp3 may not be efficiently induced in gut-resident $T_H17$ cells. Furthermore, the abundance of IL-17$^+$CD4$^+$ T cells in the LNs and spleens at steady state was also comparable in *Cd4$^{Cre}$Junb$^{fl/fl}$* and control mice (Supplementary Fig. 7b). Thus, JunB is likely dispensable for generation of a subset of $T_H17$ cells residing in the SI LP or peripheral lymphoid organs at steady state.

**JunB is required for pathogenic $T_H17$ generation *in vivo*.** To determine the *in vivo* function of JunB in pathogenicity of $T_H17$ cells, we used an EAE model, in which IL-23-dependent $T_H17$ cells have a central pathogenic function[17,18]. We immunized *Cd4$^{Cre}$Junb$^{fl/fl}$* mice with myelin oligodendrocyte glycoprotein (MOG)35–55 peptide in complete Freund's adjuvant. In contrast to control mice, which developed severe EAE, *Cd4$^{Cre}$Junb$^{fl/fl}$* mice were completely resistant to the disease (Fig. 6a). The number of CD4$^+$ T cells infiltrated into the central nervous system (CNS) was much lower in *Cd4$^{Cre}$Junb$^{fl/fl}$* mice than in control mice (Fig. 6b and Supplementary Fig. 7c). Furthermore, in the *Cd4$^{Cre}$Junb$^{fl/fl}$* mice, there were almost no IL-17A-expressing CD4$^+$ T cells and few IFN-γ-expressing CD4$^+$ T cells in the CNS (Fig. 6c and Supplementary Fig. 7c). In addition, loss of JunB severely impaired production of IL-17A-expressing CD4$^+$ T cells in LNs and spleens on day 14 after immunization (Fig. 6c

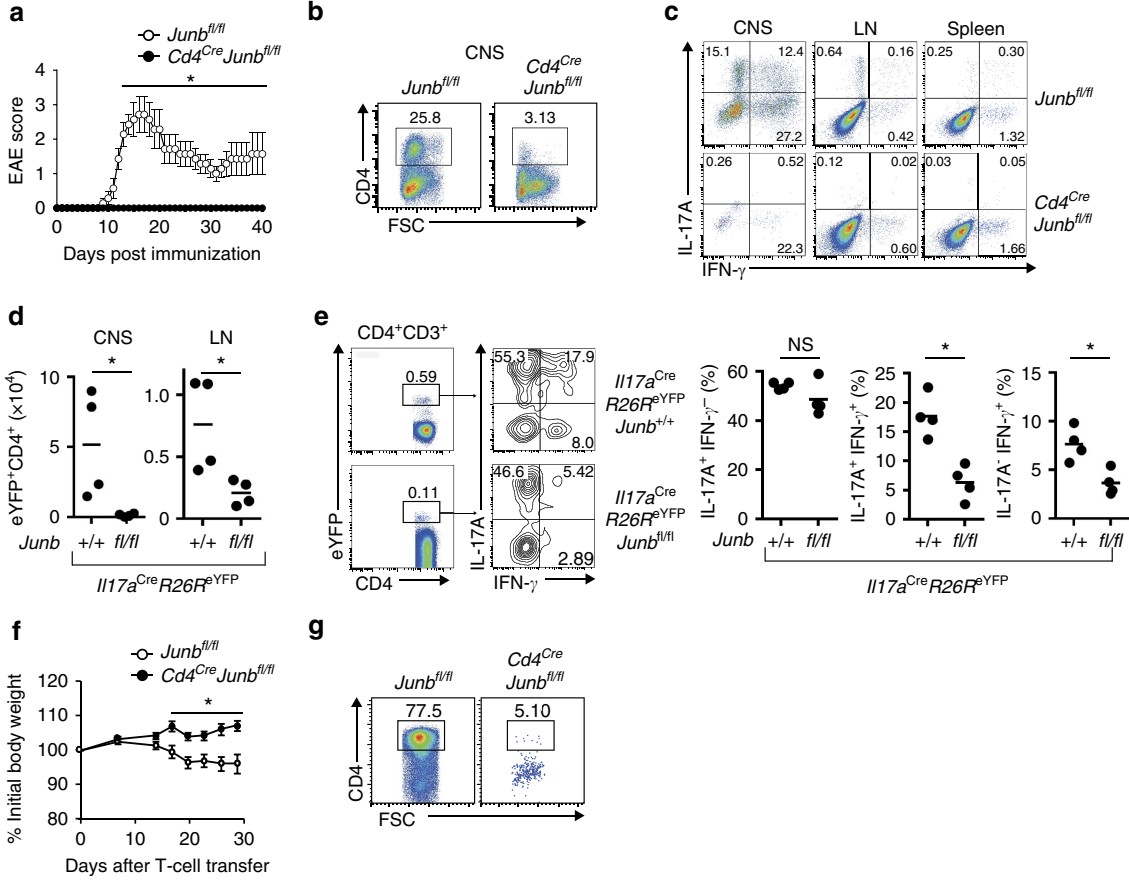

**Figure 6 | JunB is critical for T$_H$17-dependent diseases.** (**a–c**) EAE was induced in $Cd4^{Cre}Junb^{fl/fl}$ and control $Junb^{fl/fl}$ mice by immunizing with MOG peptides. (**a**) Disease development was monitored ($n = 6$). Error bars indicate s.e.m. (**b,c**) Frequencies of CD4$^+$ cells infiltrated in the CNS (**b**) and CD4$^+$ cells expressing IL-17A and IFN-γ in the CNS, spleens and LNs (**c**) were analysed 14 days after EAE induction. (**d,e**) $Il17a^{Cre}R26R^{eYFP}Junb^{fl/fl}$ or control $Il17a^{Cre}R26R^{eYFP}Junb^{+/+}$ mice were immunized with MOG peptides and cells were analysed by flow cytometry on day 14 after immunization. (**d**) Absolute numbers of CD4$^+$eYFP$^+$ cells in LNs and CNS. (**e**) Frequencies of CD4$^+$eYFP$^+$ cells expressing IL-17A and IFN-γ in LNs. (**f,g**) Colitis was induced in $Rag1$-deficient mice by transferring CD4$^+$CD45RB$^{hi}$CD25$^-$ cells from $Cd4^{Cre}Junb^{fl/fl}$ ($n = 8$) and control $Junb^{fl/fl}$ mice ($n = 12$). (**f**) Body weights of mice were monitored. Error bars indicate s.e.m. (**g**) Frequency of CD4$^+$ cells in the colon 28 days after transfer. Asterisks indicate significant differences ($P < 0.05$) by unpaired two-tailed Student's $t$-test. NS, not significant. (**a–c**) Data represent three independent experiments. (**d–g**) Data represent two independent experiments.

and Supplementary Fig. 7d). Thus, $Junb$-deficient T cells are incapable of inducing EAE.

To explore the function of JunB in T$_H$17 plasticity and stability, we performed T$_H$17-fate-mapping analysis. We induced EAE in $Junb$-deficient T$_H$17-fate-mapping reporter ($Il17a^{cre}R26R^{eYFP}$-$Junb^{fl/fl}$) mice, in which constitutive eYFP expression and JunB deficiency are induced in cells expressing IL-17A. Our data showed that abundance of eYFP$^+$CD4$^+$ T cells in the LNs and CNS in $Junb$-deficient reporter mice was significantly lower than in controls ($Il17a^{cre}R26R^{eYFP}Junb^{+/+}$) on day 14 after immunization with MOG35–55 peptide (Fig. 6d and Supplementary Fig. 7e). Furthermore, in the eYFP$^+$CD4$^+$ T-cell population, $Junb$ deficiency significantly reduced IL-17A/IFN-γ double-producing cells and IFN-γ single-producing cells, but had almost no effect on IL-17A single-producing cells (Fig. 6e), suggesting that defects in generation of T$_H$17 cells observed in $Junb$-deficient mice might not be due to increased plasticity of T$_H$17 cells. Rather, in an inflammatory context, JunB is likely required for generation of T$_H$17 cells that are competent to differentiate to IFN-γ-producing cells.

To assess whether JunB is also required for other T$_H$17-mediated diseases, we investigated the function of JunB

in colitis development by transferring $Junb$-deficient CD4$^+$ CD45RB$^{hi}$CD25$^-$ T cells into $Rag1$-deficient mice. The transfer of control, but not $Junb$-deficient CD4$^+$ T cells, induced severe weight loss, beginning about 2 weeks after the transfer (Fig. 6f). Moreover, in the colonic LP of recipient mice, the frequency of IL-17A-expressing cells in the transferred $Cd4^{Cre}Junb^{fl/fl}$ T cells was significantly lower than of controls (Fig. 6g and Supplementary Fig. 8a). However, a substantial number of the transferred $Junb$-deficient CD4$^+$ T cells expressed IFN-γ in LNs of recipient mice (Supplementary Fig. 8b).

We also investigated the function of JunB in an anti-CD3 antibody treatment model in which T$_H$17 cells are generated and migrate into the gut[44]. Injection of anti-CD3 antibody increased CD4$^+$ T cells expressing IL-17A and RORγt in the LP of duodenum in both $Cd4^{Cre}Junb^{fl/fl}$ and control mice, but the abundance of these cells was much lower in $Cd4^{Cre}Junb^{fl/fl}$ mice than in control mice (Supplementary Fig. 8c). This suggests that a subset of T$_H$17 cells can be generated independently of JunB even in an inflammatory setting, although JunB is required for full development of inflammatory T$_H$17 cells. Collectively, these results suggest that JunB may be selectively required for pathogenicity of T$_H$17 cells $in$ $vivo$.

 

## Discussion

Recent studies have revealed functionally distinct subsets within a $T_H17$ population, distinguishable by their pathogenicity and gene expression profiles[1–3]. TGF-β1/IL-6-induced $T_H17(β)$ cells are non-pathogenic, but IL-6/IL-1β/IL-23-induced $T_H17(23)$ cells are pathogenic[13,14]. Furthermore, IL-23 facilitates pathogenicity of $T_H17(β)$ cells[14]. Although a number of genes are differentially induced between $T_H17(β)$ and $T_H17(23)$ subsets, which probably contribute to their different pathogenicity, differentiation of both $T_H17(β)$ and $T_H17(23)$ subsets relies on a common $T_H17$ transcription programme composed of BATF, IRF4, STAT3 and RORγt[28–32]. However, the mechanisms of RORγt induction in TGF-β1-dependent $T_H17$ cells and IL-23-dependent $T_H17$ cells have not been fully understood.

Here we demonstrate that induction of RORγt is differentially regulated in TGF-β1-dependent $T_H17$ cells and IL-23-dependent $T_H17$ cells. Following the RNAi screen of transcription factors responsible for IL-23 signalling, we found that JunB is critical for IL-23-dependent $T_H17$ generation. Our *in vitro* results indicate that JunB is essential for RORγt induction in IL-23-dependent $T_H17$ cells, but not in TGF-β1-dependent $T_H17$ cells. Our *in vivo* results also show a similar selective requirement of JunB in a subset of $T_H17$ cells. JunB is likely dispensable for generation of gut-resident non-pathogenic $T_H17$ cells at steady state. However, JunB is required for generation of pathogenic $T_H17$ cells in EAE and colitis models. A similar phenotype has been observed in mice deficient in IL-23 signalling[15,17,19], supporting a model in which JunB facilitates IL-23-dependent pathogenic $T_H17$ production.

JunB is critical for DNA binding of BATF, IRF4 and STAT3 at the *Rorc* locus under $T_H17(23)$-polarizing conditions, but not under $T_H17(β)$-polarizing conditions. However, loss of JunB has little effect on DNA binding of these transcription factors at the *Ctla4* locus, expression of which is regulated independently of JunB even in $T_H17(23)$ conditions. These results suggest that JunB is likely required for recruitment or stable DNA binding of BATF, IRF4 and STAT3 in a target site-dependent manner under $T_H17(23)$-polarizing conditions. A heterodimer of BATF and JunB, in conjunction with IRF4, binds to AP-1-IRF composite elements in many genes, including *Rorc* and *Il17a* (refs 33,34). However, other Jun proteins, c-Jun and JunD, also interact with BATF/IRF4 and bind to AP-1-IRF composite elements[33,34]. Furthermore, RNAi ablation of c-Jun impairs TGF-β1-dependent $T_H17$ differentiation[45]. These findings suggest that DNA-binding activity of BATF may be controlled by its AP-1 heterodimer partners, including JunB and c-Jun, in a target site- and context-dependent manner.

IL-23 is required for maturation and maintenance of $T_H17$ cells, which are likely linked to acquisition of $T_H17$ pathogenic cytokine-producing ability and functional plasticity in the late phase of $T_H17$ development[6]. Our data show that a subset of $T_H17$ cells is generated independently of JunB in the early phase of EAE, but full development of inflammatory $T_H17$ cells depends on JunB. This suggests that JunB may be important for IL-23-dependent maturation and maintenance of $T_H17$ cells, which are probably generated in a TGF-β-dependent manner. Consistent with this, our *in vitro* data show that JunB facilitates IL-23-dependent maintenance of IL-17-producing ability of TGF-β/IL-6-induced $T_H17$ cells. Furthermore, *in vivo* $T_H17$-fate-mapping data indicate that JunB is important for generation of $T_H17$-derived IFN-γ-producing cells. Seemingly contradicting the *in vivo* observation, however, JunB deficiency results in abnormal induction of T-bet and IFN-γ in $T_H17$ cells *in vitro*. In addition, our ChIP-seq results show that JunB, together with BATF, IRF4 and STAT3, binds to the *Tbx21* locus, suggesting direct regulation of T-bet expression by JunB in $T_H17$

cells. Collectively, these data suggest that JunB may control $T_H17$ plasticity in part by positively or negatively regulating T-bet expression in $T_H17$ cells in a developmental-phase-dependent manner, probably by interacting with distinct transcription factors or epigenetic regulators.

RORγt regulates induction of a restricted subset of $T_H17$ genes[29]. Consistent with this, JunB deficiency impairs expression of a small number of genes, such as *Il17a*, *Il17f* and *Il23r* under $T_H17(23)$-polarizing conditions. Despite impaired induction of a limited number of $T_H17$ genes, *Junb*-deficient T cells lose their ability to induce EAE and colitis, implying a critical function of these JunB-regulated genes in $T_H17$ pathogenicity. As *Il-23r*-deficient T cells are also incapable of inducing $T_H17$-dependent diseases[17], JunB-dependent IL-23R induction is likely critical for pathogenicity of $T_H17$ cells. Furthermore, given that IL-23 signalling is important for RORγt expression in the absence of TGF-β1, it is possible that the diminished RORγt expression in *Junb*-deficient $T_H17(23)$ cells might be due to impaired IL-23R induction, which is needed for full induction of RORγt.

It has been suggested that GM-CSF and IFN-γ are involved in $T_H17$ pathogenicity[26,27,46]. However, JunB is not required for induction of GM-CSF and IFN-γ under $T_H17(23)$-polarizing conditions. These data suggest that GM-CSF and IFN-γ need to be produced by JunB-dependent $T_H17$ cells to exert their pathogenic functions. A recent report demonstrated that Blimp1, which is induced by IL-23 *in vivo*, is critical for pathogenic $T_H17$ differentiation[42]. Interestingly, our ChIP-seq data showed that JunB and BATF/IRF4 bind to the *Prdm1* locus in $T_H17$ cells *in vitro*, suggesting that JunB may have a function in regulation of Blimp1 expression at late stages of $T_H17$ differentiation. We also found that JunB binds to the *Cd5l* locus in $T_H17(23)$ cells, but not in $T_H17(β)$ cells. CD5L is induced in non-pathogenic $T_H17$ cells and inhibits their pathogenicity[43]. It is intriguing to speculate that JunB-dependent negative regulation of CD5L production may be involved in regulation of pathogenicity of $T_H17$ cells.

Although JunB is required for RORγt induction specifically in $T_H17(23)$ cells, another pathway that is activated by TGF-β1 may compensate for the absence of JunB. Although the main TGF-β1 signalling pathway is mediated by SMAD2, SMAD3 and SMAD4 transcription factors, previous data have shown that the SMAD-dependent pathway is unnecessary for TGF-β1-dependent RORγt induction[47]. Our preliminary analysis showed that TGF-β1 receptor kinase inhibitor (SB43152) treatment significantly inhibited *Rorc* induction in $T_H17(β)$ cells, whereas there was only a partial or no reduction of *Rorc* expression in $T_H17(β)$ cells treated with TGF-β1 signalling regulators, including JNK inhibitor (SP600125), MEK inhibitor (PD98059), p38 inhibitor (SB203580), PI3 kinase inhibitor (LY294002), SMAD3 inhibitor (SIS3) or ROCK inhibitor (Y27632) (Supplementary Fig. 9). Thus, TGF-β receptor kinase activity may be important for JunB-independent $T_H17$ differentiation, but the downstream signalling pathways remain unknown.

In conclusion, we demonstrate that JunB is selectively required for activation of the $T_H17$ core transcription programme in IL-23-dependent pathogenic $T_H17$ cells, but that the JunB-independent pathway is sufficient to activate the same programme in TGF-β1-dependent, non-pathogenic $T_H17$ cells. JunB is essential for expression of RORγt, by promoting DNA binding of BATF, IRF4 and STAT3 at the *Rorc* locus in IL-23-dependent pathogenic $T_H17$ cells. *Junb*-deficient T cells are incapable of inducing EAE and colitis, but loss of JunB does not seem to affect $T_H17$ generation in the gut at steady state. Thus, the JunB-dependent pathway could be an attractive therapeutic

 

target to suppress pathogenicity of $T_H17$ cells, while maintaining beneficial $T_H17$ populations.

## Methods

**Mice.** To generate T-cell-specific, *Junb*-conditional knockout mice, we used C57BL/6-background ES cells, EGR-1, carrying a 'knockout first' *Junb* tm1a allele (Eucomm)[48], which contains flippase recombination target (FRT)-flanked *lacZ* and neomycin resistance (*neo*) cassettes in front of a loxP-flanked (floxed) *Junb* exon 1 (Supplementary Fig. 2a). ES cells were injected into eight-cell Institute for Cancer Research (ICR) mouse embryos, and chimeric blastocysts were transferred into the uteri of pseudo-pregnant ICR female mice[49]. The resultant chimeric mice were crossed with *FLP* mice (Jackson, Stock No. 009086) to excise the FRT-flanked region, which generated mice carrying a conditional *Junb* tm1c allele (floxed *Junb* mice) (Supplementary Fig. 2a). Floxed *Junb* mice were crossed with *Cd4^Cre* mice (Jackson) to create *Cd4^Cre Junb^{fl/fl}* mice in which T cells carry deleted *Junb* (tm1d) alleles (Supplementary Fig. 2a). *Il17a^cre* (Stock No. 016879) and *Rosa26^eYFP* (*R26R^eYFP*) (Stock No. 006148) mice were from the Jackson Laboratory. C57BL/6 mice and floxed *Stat3* mice[36] were purchased from CLEA Japan and Oriental Bioservice, respectively. All mice were housed under specific pathogen-free conditions. Gender-matched 6–12-week old mice were used for experiments. All animal experiments were performed following protocols approved by Animal Care and Use Committee at Okinawa Institute of Science and Technology Graduate University.

**Antibodies.** The following antibodies were used for flow cytometry analysis and fluorescence-activated cell sorting (FACS): Anti-CD3 (17A2, Biolegend, 1:400), anti-CD4 (GK1.5, Biolegend, 1:100 or 1:400), anti-CD8 (53–6.7, Biolegend, 1:400), anti-CD25 (PC61, Biolegend, 1:400), anti-CD44 (IM7, Biolegend, 1:400), anti-CD62L (MEL-14, Biolegend, 1:400), anti-IL-17A (TC11-18H10.1, Biolegend, 1:100), anti-IFN-γ (XMG1.2, Biolegend, 1:100), anti-GM-CSF (MP1-22E9, BD, 1:100), anti-RORγt (Q31-378, BD, 1:50), anti-Foxp3 (150D, Biolegend, 1:50), anti-T-bet (4B10, Biolegend, 1:50) and anti-TCRVβ8.1/8.2 (KJ16-133.18, Biolegend, 1:20). For immunoblot analyses, anti-JunB (C37F9, Cell Signaling Technology, 1:2,000), anti-RORγt (AFKJS-9, eBioscience, 1:400) and anti-GAPDH (3H12, MBL, 1:2,000) were used. For ChIP analyses, anti-JunB (210, Santa Cruz, 2 µg per ChIP), anti-BATF (WW8, Santa Cruz, 2 µg per ChIP), anti-IRF4 (M-17, Santa Cruz, 2 µg per ChIP), anti-STAT3 (c-20, Santa Cruz, 2 µg per ChIP) were used.

***In vitro* CD4$^+$ T-cell differentiation.** CD4$^+$ T cells from single-cell suspensions of murine spleens and LNs were enriched using a MACS magnetic cell sorting system with anti-CD4 microbeads (130-049-201, Miltenyi). Then naive CD4$^+$ T cells (CD4$^+$CD25$^-$CD62L$^{hi}$CD44$^{lo}$) were sorted by FACS AriaII or AriaIII (BD). Cells were activated with plate-bound anti-CD3 antibody (5 µg ml$^{-1}$; 145-2C11, Biolegend) and soluble anti-CD28 antibody (1 µg ml$^{-1}$, 37.51, Biolegend) in IMDM media (Invitrogen) supplemented with 10% foetal calf serum (Invitrogen) containing the following cytokines and antibodies: IL-2 (20 ng ml$^{-1}$; Biolegend), anti-IFN-γ (1 µg ml$^{-1}$; Biolegend) and anti-IL-4 (1 µg ml$^{-1}$; Biolegend) for $T_H0$; IL-6 (20 ng ml$^{-1}$; Biolegend) and TGF-β1 (3 ng ml$^{-1}$; Miltenyi) for $T_H17(\beta)$; IL-6 (20 ng ml$^{-1}$; Biolegend), IL-1β (20 ng ml$^{-1}$; Bioelegend) and IL-23 (40 ng ml$^{-1}$; Biolegend) for $T_H17(23)$; IL-6 (20 ng ml$^{-1}$) and TGF-β3 (3 ng ml$^{-1}$; Miltenyi) for $T_H17(\beta3)$;TGF-β1 (15 ng ml$^{-1}$) and IL-2 (20 ng ml$^{-1}$) for iTreg differentiation. For analysis of cytokine expression, cells were re-stimulated with phorbol 12-myristate 13-acetate (PMA; 50 ng ml$^{-1}$; Sigma) and ionomycin (500 ng ml$^{-1}$; Sigma) in the presence of brefeldin A (5 µg ml$^{-1}$; Biolegend) for 4 h. Then, cells were fixed with 4% paraformaldehyde, permeabilized in permeabilization/wash buffer (421002, Biolegend), and stained with antibodies against cytokines. For analysis of expression of RORγt and Foxp3, Foxp3 staining buffer set (00-5253-00, eBioscience) was used according to the instructions. In some experiments, TGF-β1 receptor kinase inhibitor (SB43152, 10 µM; Selleckchem), JNK inhibitor (SP600125, 10 µM; EMD Millipore), MEK inhibitor (PD98059, 10 µM; Invivogen), p38 inhibitor (SB203580, 10 µM; Invivogen), PI3 kinase inhibitor (LY294002, 5 µM; Invivogen), SMAD3 inhibitor (SIS3, 10 µM; Sigma) or ROCK inhibitor (Y27632, 10 µM; Sigma) were added to the culture media. Dead cells were excluded using Zombie NIR Fixable viability kits (423106, Biolegend) for flow cytometry analysis. All flow cytometry gating strategies are shown in Supplementary Fig. 10.

**Cell isolation.** Mononuclear cells were isolated from SI LP or colonic LP using Lamina Propria Dissociation kits (130-397-410, Miltenyi). For isolation of cells from CNS of EAE-induced mice, brains and spinal cords were digested with collagenase D (1 mg ml$^{-1}$; Roche) and DNase I (2.5 mg ml$^{-1}$; Sigma) in phosphate-buffered saline at 37 °C for 1 h with shaking. Then, isolated cells were re-suspended in 37% Percoll (GE Healthcare), mixed with 70% Percoll and then centrifuged for 20 min. Mononuclear cells were isolated from the interface. Cells were incubated with anti-Fc receptor blocking antibody (anti-CD16/CD32; Biolegend) and then stained with antibodies against cell surface molecules. Expression of cytokines and transcription factors was analysed by flow cytometry as described above.

**Enrichment of IL-17-secreting cells.** We enriched IL-17-secreting cells with a mouse IL-17 secretion assay kit (Miltenyi), according to the manufacturer's instructions. Briefly, naive CD4 T cells were activated in the presence of TGF-β1 and IL-6, as above. On day 3, cells were re-stimulated with PMA (10 ng ml$^{-1}$) and ionomycin (1 µg ml$^{-1}$) for 3 h, followed by IL-17-capture reaction in serum-free media (X-VIVO20; Lonza). Phycoerythrin-labelled IL-17-secreting cells were sorted using a FACS AriaIII.

**Retrovirus infection.** The pMKO1.GFP retroviral vector (a gift from William Hahn, Addgene plasmid # 10676), which is a bicistronic vector containing an internal ribosome entry site (IRES). IRES-driven complementary DNA (cDNA) encoding green fluorescent protein (GFP) was used for shRNA transduction. Briefly, PlatE cells were transfected with pMKO1.GFP containing shRNAs and the pCL-Eco helper plasmid using polyethylenimine. pCL-Eco was a gift from Inder Verma (Addgene plasmid # 12371)[50]. Culture supernatants were collected at 72 h post transfection, supplemented with polybrene (8 µg ml$^{-1}$), and added to sorted naive CD4$^+$ T cells (2 × 10$^5$ cells per well in a 48-well plate) previously stimulated for 36 h under $T_H0$-polarization conditions. Cultures were centrifuged at 300g for 60 min at room temperature, and media was replaced with fresh media containing $T_H17(23)$ cytokines. Cells were incubated for 3 days, and GFP$^+$ cells were sorted for RNA isolation and qRT–PCR. Sequences of shRNAs against 263 transcription factors highly expressed in $T_H17$ cells are listed in Supplementary Data 1.

**qRT–PCR.** Total RNA isolated from cells using an RNeasy Plus Mini Kit (74136, Qiagen) was used for cDNA synthesis with a Revertra Ace qPCR Kit (FSQ-101, Toyobo). The resulting cDNA was used as a template for qRT–PCR performed with Faststart SYBR master mix (4673484, Roche) and a Thermal Cycler Dice Real Time system (Takara). Primers used for qPCR are listed in Supplementary Table 1.

**Immunoblot analysis.** Cells were lysed with RIPA buffer (Thermo) containing complete protease inhibitor cocktail (4693159, Roche). Cellular debris in the lysate was removed by centrifugation at 14,000g for 15 min. The protein concentration was measured with a DC Protein Assay kit (500-0106, Bio-Rad). Protein extracts were mixed with 5 × sample loading buffer (250 mM Tris-HCl pH 6.8, 10% SDS, 30% glycerol, 5% β-mercaptoethanol and 0.02% bromophenol blue) and subjected to SDS–polyacrylamide gel electrophoresis. Then separated proteins on the gel were transferred onto Immobilon P transfer membranes (Millipore) using a Trans-blot electrophoretic transfer system (Bio-Rad). Membranes were blocked with 5% skim milk (Wako) in phosphate-buffered saline with 0.1% Tween-20 (Sigma) and incubated with the indicated primary antibodies and probed with horseradish peroxidase-conjugated anti-mouse or anti-rabbit IgG antibodies (Cell Signaling Technology, 1:4,000). Bound antibody was detected with Clarity Western ECL (Bio-Rad) or SuperSignal West Femto detection reagents (Thermo) and an Las-3000 imaging system (Fuji film). Uncropped original scans of immunoblots were provided in Supplementary Fig. 11.

**Microarray analysis.** Total RNA was isolated from cells using an RNeasy Plus Mini Kit (74136, Qiagen). RNA quality was analysed with an Agilent 2100 Bioanalyzer and an RNA 6000 Nano Kit (5067-1511, Agilent). RNA samples showing RNA integrity numbers ≥7 were used to generate biotinylated complementary RNA (cRNA) with a GeneChip WT Plus Reagent Kit (902280, Affymetrix) and GeneChip Hybridization, Wash and Stain Kit (900720, Affymetrix). Labelled cRNA was hybridized to Affymetrix Mouse Gene 1.0 ST microarrays (Affymetrix), then stained and scanned with an Affymetrix GeneChip 3000 (Affymetrix).

**ChIP-seq and ChIP–PCR analysis.** ChIP was performed using a SimpleChIP Plus Enzymatic Chromatin IP Kit (9005S, Cell Signaling) with some modifications, as previously described[51,52]. Briefly, activated T cells (10$^6$ per ChIP-seq or 1–3 × 10$^5$ per ChIP–PCR) were crosslinked in culture medium containing 1% formaldehyde at room temperature for 10 min, and the reaction was stopped by adding glycine solution. Then cells were lysed, and nuclei were collected and treated with micrococcal nuclease (0.0125 µl ml$^{-1}$) for 20 min at 37 °C. After stopping the reaction with 0.05 M EGTA, samples were sonicated with several pulses to disrupt nuclear membranes. Then the supernatant, containing chromatin, was collected after centrifugation. Chromatin solutions were incubated with 2 µg of antibodies overnight at 4 °C with rotation, followed by incubation with Protein G magnetic beads for 2 h at 4 °C. Beads were washed, and chromatin was eluted. Crosslinks were reverted according to kit instructions. DNA was purified by phenol/chloroform extraction and used for ChIP–PCR analysis with primers listed in Supplementary Table 1.

For ChIP-seq analysis, sample DNA was quantified with a Qubit 3.0 Fluorometer (Thermo Fisher Scientific) and normalized to 200 pg as starting DNA for library preparation. Libraries were prepared using KAPA Hyper Prep Kit (KK8500, KAPA Biosystems) protocols for blunt-ending, polyA extension and adaptor ligation. Post-ligation clean-up with an Agencort AMPure XP (Beckman Coulter) was performed at a 1.8 × DNA ratio to purify ligated DNA, which was then PCR-amplified and purified using AMPure XP at a 1.2 × DNA ratio to

remove excess adaptor-dimer and to preserve small fragments. Size selection was performed with a 2% agarose gel cassette of Blue Pippin (Sage Science) for a target insert size between 30 and 180 bp. Library quantification was performed by droplet digital PCR (Bio-Rad). All libraries were pooled and loaded onto cBot (50 μl of 200 pM) for cluster generation, and then sequenced on an Illumina HiSeq4000 at a target sequencing depth of 10 million uniquely aligned reads.

**EAE induction.** Six–eight-week old, gender-matched mice were immunized with MOG35–55 peptides (300 μg per mouse) in complete Freund's adjuvant (CFA) (100 μl per mouse) containing dead *Mycobacterium tuberculosis* (1 mg per mouse). Pertussis toxin (400 ng per mouse) was also intraperitoneally injected into the mice twice on day 0 and on day 2 post immunization. Disease severity was evaluated on a scale of 1–5 as follows: 1, limp tail; 2, partially paralysed hind legs; 3, completely paralysed hind legs; 4, complete hind and partial front leg paralysis; 5, completely paralysed hind and front legs. Mice with disease score 5 were considered moribund and were killed by $CO_2$ inhalation.

**Colitis induction.** $CD4^+CD45RB^{hi}CD25^-$ T cells were purified from $CD4^+$ T cells isolated from spleens and LNs of $Cd4^{Cre}Junb^{fl/fl}$ and control mice, and intraperitoneally injected into $Rag1$-deficient mice ($4 \times 10^5$ cells per mouse). Disease progress was monitored by weighing the mice.

**Anti-CD3 antibody treatment.** We injected mice with anti-CD3 antibody (50 μg per mouse) three times at 0, 48 and 96 h. At 4 h after the final injection, the mice were killed by $CO_2$ inhalation for cell isolation from LP of duodenums.

**Statistical analysis.** Statistical analyses were performed using unpaired two-tailed Student's $t$-test with Prism software (GraphPad). $P$ values $<0.05$ were considered as significant.

**Data availability.** Microarray and ChIP-seq data that support the findings of this study have been deposited in the Gene Expression Omnibus with the primary accession codes GSE86499 and GSE86535, respectively. The authors declare that all other data supporting the findings of this study are available within the article and its Supplementary Information files or are available from the authors on request.

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

## Acknowledgements

We thank M. Matmati, B. Parajuri, S. Nishijima, H. Goto and M. Igawa for discussion and assistance, S. Akira for floxed *Stat3* mice, C. Wilson for *Cd4^Cre^* mice, W. Haln and I. Verma for plasmid constructs and S. Aird for editing the manuscript. We also thank NPO Biotechnology Research and Development for technical assistance. We are also grateful to OIST Graduate University for its generous funding of the Immune Signal Unit.

## Author contributions

Z.H., S.-i.K., D.S., H.Y., S.O. and H.S. conducted experiments. Z.H., S.-i.K., and D.S. analysed data. N.A. carried out ChIP-seq experiments. Y.F. generated floxed JunB mice. H.I. designed experiments and wrote the manuscript.

## Additional information

**Competing interests:** The authors declare no competing financial interests.

