## [Peer Review File · Nature Communications]

Reviewers' comments:

Reviewer #1 (Remarks to the Author):

In this manuscript, Hasan et al analyse the transcriptional program used by pathogenic Th17 cells raised in the presence of IL-23 as compared to conventional Th17 cells raised by TGF β plus IL-6. The authors discover a particular role for JUNB in pathogenic Th17 cells. Ablation of JUNB leads to cells lacking key molecules necessary for Th17 function and this is correlated to reduced binding of IRF4 and BATF to relevant genetic loci in the absence of JUNB. The relevance of these findings for diseases in vivo is shown in the EAE and colitis model.

I find these data relevant and in principle worth to be published by Nat. Commun. However, I have several issues which should be addressed before:

Major points:

So, if JUNB is less important in Th17 (b) cells, can the authors find evidence for another transcription factor which may be able to replace JUNB in these cells? In particular, as cited, it has been shown that knock-down of c-JUN affects differentiation of TGF β -induced Th17 cells, which according to the present nomenclature, would be non-pathogenic Th17 cells. How about a role of c-JUN for pathogenic Th17 cells?

An issue is whether JUNB ablation locks expansion of any Th17 cells in vivo, not just pathogenic ones. The results obtained in the periphery during EAE and colitis suggest this possibility. Can the authors test expansion of non-pathogenic Th17 cells in a situation in which the described pathogenic Th17 cells are less important, perhaps a bacterial infection – or e.g. the response of gut Th17 cells to SFB, the Th17 cell inducing bacteria detected by the Littmann group?

I miss further analysis of the Th17 cells which are induced by IL-6 plus TGF β 3 which were also shown previously to be pathogenic, even independently of T-bet (Lee et al, NI, 2012). How is their transcriptional regulation compared to the Th17 (23) cells?

Minor points: In Fig. 2b, the staining for ROR γ T is not convincing. Can this finding be repeated by a Western Blot?

In sFig. 1b: Was this result obtained using Th17 conditions?

Reviewer #2 (Remarks to the Author):

In this manuscript the authors investigated the function of the transcription factor JunB in the development of Th17 cells both in vitro and in vivo. Based on comprehensive analysis they propose the following model: IL-6 stimulates the expression of JunB, which promotes the expression of ROR γ T specifically in IL-23-dependent pathogenic Th17 cells. This model is of great potential interest if authors will propose which factor(s) drive the expression of ROR γ T in non-pathogenic Th17 cells (instead of the IL-6/JunB axis), the cell type abundantly found in the gut of JunB deficient mice and in the in vitro cultures. The intriguing fact is that Th17 cell development in non-pathogenic conditions is also driven by the same set of transcription factors (namely IRF4, BATF and STAT3), activity of which is proposed here to be JUNB-dependent. Could it be IL-21? Or different sources of IL-6 are important? A main problem I see is that all conditions tested here included IL-6, which ultimately stimulated the expression of JunB and ROR γ T (not everywhere shown), leading to the generation of Th17 cells. However, effects of additionally added cytokines to mimic pathogenic (+ IL-1 β + IL-23) and non-pathogenic (+ TGF β 1) conditions could be independent of JunB and do not allow to put IL-23 as a key signaling partner of JunB without additional experimental support. What would happen if IL-23 is

added into non-pathogenic (IL-6/TGF β 1) conditions? Will it mimic effects found in IL-6/IL-1 β /IL-23 conditions?

Apart of aforementioned concern and several minor points, which are listed below; the study is well structured and written. Overall the results are solid, convincing and shed significant light in our understanding of Th17-cell mediated pathology.

Comments:

1) Suppl Fig 1. Authors used Enpp2 mRNA levels as an indicator of Th17 cells development. Not clear why they choose this marker over broadly accepted Th17-defining products such as Rorc, IL-17 etc.? The fact that knockdown of JunB on Th17 cells resulted in reduction of Enpp2 levels 3 days after transfection speaks more in favor of the maintenance requirements, rather than induction of Th17 cell development as they claim (Suppl Fig 1b).

2) Fig 1. Since IL-6 alone is enough to induce strong JunB expression and neither IL-1 nor IL-23 increased this effect, how IL-23 promotes pathogenic Th17 cell development in an JunB-dependent manner as authors claim? Authors need to provide more evidences to distinguish IL-6/IL-23 discrepancy.

3) Fig 2a-d. Data presented here show that IL-6 fail to suppress Tregs and Th1 development in JunB deficient background. Does JunB regulate IL-6R expression or it serves as direct suppressor of Foxp3 and T-bet/Eomes? What is the relevance of discovered strong ROR γ t-Foxp3 co-expression without IL-17 production here and in in vivo system? FACS plots should be supplemented with total cell numbers in order to support the claim of developmental defects and not increased plasticity of JunB deficient Th17 cells towards other lineages. Changing of X- and Y-axis on FACS plots (panel b) will make easier the comparison between IL-17A and ROR γ t expression (the same is for the panel d).

4) Fig 2e. Authors claim that JunB deficient Th17 cells generated in non-pathogenic conditions loose their IL-17 expression after re-stimulation with IL-23. This statement requires additional clarification. What happened to the initial Th17(β) cells, they died or changed their cytokine profile? In other words, do abundant IFN γ expression found in JunB deficient cultures supplemented with IL-23 originated from exTh17 cells? Analysis of cell numbers may help in answering this question if authors do not have Th17-fate mapping system in present.

5) Fig 3. Does IL-10 expression dependent on JunB? Please discuss this point.

6) Fig 4d. Authors state: "Loss of JUNB considerably diminished DNA-binding of BATF, IRF4, and STAT3 at the Rorc locus, under TH17(23) conditions, but not under TH17(β) conditions (Fig. 4d)." The corresponding figure shows no changes under Th17(β) conditions only for BATF, while IRF4 and STAT3 input indicated to be significantly decreased. Explain this discrepancy.

7) Fig 5. As a suggestion, authors may want to use an anti-CD3 treatment model, which is proposed to be strongly dependent on IL-6/TGF β to strengthen non-pathogenic Th17 cells development in in vivo system.

8) Fig 6. Authors need to show numbers of CNS infiltrating CD4 T cells and especially Th1 cells (shown in panel c), to support developmental failure over the migration disadvantage or aberrant plasticity.

General comments:

In many in vivo experiments the gating strategy is not clear. Did authors use any markers to define T cells? Gating on only CD4 positive cells seems to be inappropriate, especially for the gut samples normally enriched with CD4+ ILC's.

Authors should state the JunB expression levels in a novel JunB conditional KO mice, used as a controls, in comparison to C57BL/6 wild type mice.

Standardize the gene symbols used in the methods part.

In the introduction: "Furthermore, TGF- β 1 diminishes pathogenicity of TH17 cells isolated from EAE-induced mice, while IL-23 makes TH17(β) cells pathogenic, suggesting that TGF- β 1 and IL-23 signals facilitate differentiation of non-pathogenic and pathogenic TH17 cells, respectively."

Please re-write. It is broadly accepted that IL-23 is needed for Th17 cell pathogenicity and can convert non-pathogenic Th17 cells into pathogenic.

In the discussion: "ROR γ t regulates induction of a small subset of TH17 genes."

Word restricted or certain seems to be more appropriate in this context.

Reviewer #3 (Remarks to the Author):

This manuscript by Hasan et al revisits a currently much debated issue focused on transcriptional control of pathogenic vs non-pathogenic Th17 cells. There has been a flurry of papers in the last couple of years attempting to identify markers of such functional states in Th17 cells with a view of pinpointing potential therapeutic targets in pathogenic Th17 responses.

As such the present manuscript is not particularly original, but adds another facet to the ongoing debate. Similar to previous approaches the authors focus on in vitro generated Th17 cells using two different protocols that are reported to generate either pathogenic or non-pathogenic Th17 cells. There are numerous permutations for in vitro generation of Th17 cells with many associated deficiencies and none of them fully reflect the Th17 effector state in vivo. A particular problem for these approaches is the heterogeneous nature of the cells growing out of these cultures and in the absence of a reporter system it is simply not clear how many of them are actually Th17 cells.

The major claims in the manuscript are:

JUNB is essential for induction of ROR γ t via facilitating binding of BATF, IRF4 and STAT-2 in IL-23 dependent but not TGF β dependent Th17 cells. There is a selective requirement for JUNB for pathogenic Th17 differentiation.

It is possible though that the effects of JUNB are related to plasticity vs stability rather than pathogenicity or non-pathogenicity. It would be preferable to use a reporter or better even fate reporter to address these issues and avoid the in vitro protocols which are not without controversy. It is quite evident from Fig.2 that absence of JUNB allows the emergence of Tregs and Th1 in vitro. So does JUNB regulate ROR γ t induction or rather restrain Treg/Th1 differentiation? ROR γ t staining in Fig.2b (Th17(23) looks essentially negative also in the control group (JUNB $^{fl/fl}$).

It is difficult to interpret the microarray results, which are based on in vitro cultures of very

heterogeneous populations between the two conditions.

The in vivo data on gut resident Th17 cells are interesting, suggesting that JUNB has no impact on their generation. However, the fact that Foxp3+ cells were reduced in the absence of JUNB is in direct contradiction to the in vitro results, suggesting that it is problematic to extrapolate from in vitro cultures of Th17 cells to in the in vivo scenario.

With respect to the EAE data in Fig.6 it is evident that a Th17 response is simply never initiated- not a scenario of non-pathogenic vs pathogenic induction. Given that JUNB deficiency seems to strongly affect expression of IL-23R (Fig.3) this is not surprising as EAE is not induced in the absence of IL-23 signals, shown initially by Dan Cua's work. Without a Th17 response that exceeds a certain threshold there is no infiltration of other cells types in the CNS, but the sequelae of this have been described before and there are many genetic modifications of the Th17 program that result in the same phenotype= lack of CNS pathology in this model.

So altogether this manuscript identified another important player in the Th17 pathway, but the attempts to link this to decisions regarding pathogenicity or not are not convincing and there is a lack of connection between observations in vitro and in vivo data that further complicate an already difficult issue.

Reviewers' comments:

Reviewer #1 (Remarks to the Author):

I think that the authors did a good job to address the raised issues. They could not clarify everything, but the topic is complicated indeed - and the new information is significant. I have only some small points left:

Line 720, 793: SD of what exactly? Of the two experiments?

Fig. 1c,d: also naive CD4+ T cells?

In suppl. Fig. 4, it is striking that JunB expression is much less in Th17 (23) than in Th17 (β) cells. This is in striking contrast to Fig. 1. Why? How often was this performed?

L 169, „not“ does not reflect the data. „Less striking“ would be better.

The legend for Fig. 3a is still quite hard to understand for me. It uses the values of suppl 5a but compares ko vs wt cells, correct? This should be clarified.

L 323 these are not Cd4CreJunBfl/fl mice, but RAG ko mice having been transplanted with Cd4CreJunBfl/fl cells??

Reviewer #2 (Remarks to the Author):

The authors need to familiarize themselves with the developmental/maintenance issue of JunB deficient cells. I agree that presented data is not firmly enough to claim its necessity for Th17 cells maintenance/plasticity, however it is a very likely additive scenario and authors need to mention it at least in the discussion part.

Moreover, in light of a new data, ALL effects of JunB deficiency seem to be a result of impaired IL-23R expression, and this needs to be clearly stated in the title and abstract. In brief, IL-6 facilitates IL-23R expression in a JunB dependent manner. To note, IL-23 is not a differentiation factor for Th17 cells, unless authors wish to challenge this paradigm.

Minor points:

1) Make clear in the text what is the Enpp2 and why it was used as a readout for Th17 cells development.

2) Supplementary Fig. 4b. From the letter: "Indeed, the Foxp3-low population expressed higher levels of IL-17A than the Foxp3-high population in JunB-deficient cells under Th17(β) conditions (Supplementary Fig. 4d)".

Not convincing. Use MFI for IL-17A staining to claim the difference. The reference to this figure in the manuscript is not appropriate.

3) Figure C supports unaffected initial IL-6 response by JunB deficient cells and can be used as supplementary figure, after explanation of the statistical significance in Th17(23) conditions.

4) Fig 2b. Such dim RORγt staining cannot be shown in a main figure.

5) Fig. 6d. Authors write in the manuscript “proportion of eYFP+CD4+ T cells in JunB-deficient reporter mice was slightly lower than controls”.

Four times decrease cannot be called a slight decrease, although could be not significant. Overall, day 7 after immunization is not the best time to investigate the plasticity by using fate-mapping system, later time points (and ideally CNS analysis) seems to be more appropriate for that purpose.

6) Supplementary Fig 7e. Did I miss the legend for this panel?

Reviewer #3 (Remarks to the Author):

The authors have made substantial efforts to answer the reviewer’s queries.

In particular the inclusion of in vivo data with the IL-17 fate reporter answered an important question and points towards a function of JunB in IL-23 mediated pathogenicity. The remaining problem I see is the disconnection with the in vitro data. Here the authors claim that JunB deficiency leads to an increase in IFN γ and GM-CSF- in direct contradiction to what the fate reporter indicated in vivo. Furthermore, the in vivo data show that the initial generation of Th17 cells that produce IL-17 is not suppressed, but the cells decline after the initial activation in line with what was found in IL-23 deficient mice.

However, in vitro JunB deficiency is claimed to compromise ROR γ t expression and thereby IL-17 with varying effects depending on the somewhat artificial culture conditions. The authors concede that there are ‘gaps’ between Th17 cells generated in vitro and in vivo, but chose not to discuss these issues in the manuscript. Using the fate reporter for the in vitro experiments would have established that the increase in IFN γ and GM-CSF comes most likely from alternative T cell fates as anti-CD3 activates any T cell depending on subsequent cytokine conditions and transcriptional programs. As the authors show ROR γ t and IL-17a defective under Th17 (23) polarizing conditions, which are equated in the literature with generation of ‘pathogenic’ Th17 cells, this does not link to the largely unperturbed Th17 cell induction in vivo, which wanes before the cells undergo IL-23 dependent plasticity towards IFN γ and/or GM-CSF secretion.

There should at least be a critical discussion of these discrepancies in the manuscript.

REVIEWERS' COMMENTS:

Reviewer #2 (Remarks to the Author):

The authors addressed my concerns satisfactorily

Reviewer #3 (Remarks to the Author):

I am satisfied with the authors response to my remaining concerns and I particularly like the change of title that clarifies several points. As far as I am concerned the manuscript is suitable for publication

Responses to reviewers' comments.

We greatly appreciate the constructive reviews provided by the reviewers. Below, please find our point-by-point responses to questions and comments. We have performed additional experiments, revised the manuscript text, and provided some new figures. Particularly, our main claim is further bolstered by new results of *JunB*-deficient Th17-fate mapping reporter experiments and by an analysis of IL-23 effects on purified *JunB*-deficient IL-17-expressing cells. The reviewers' critiques have strengthened our paper.

Reviewer #1

In this manuscript, Hasan et al analyse the transcriptional program used by pathogenic Th17 cells raised in the presence of IL-23 as compared to conventional Th17 cells raised by TGF β plus IL-6. The authors discover a particular role for JUNB in pathogenic Th17 cells. Ablation of JUNB leads to cells lacking key molecules necessary for Th17 function and this is correlated to reduced binding of IRF4 and BATF to relevant genetic loci in the absence of JUNB. The relevance of these findings for diseases in vivo is shown in the EAE and colitis model.

I find these data relevant and in principle worth to be published by Nat. Commun. However, I have several issues which should be addressed before:

- 1. So, if JUNB is less important in Th17 (b) cells, can the authors find evidence for another transcription factor which may be able to replace JUNB in these cells? In particular, as cited, it has been shown that knock-down of c-JUN affects differentiation of TGF β -induced Th17 cells, which according to the present nomenclature, would be non-pathogenic Th17 cells. How about a role of c-JUN for pathogenic Th17 cells?*

We would like to thank this reviewer for asking this critical question. We have been struggling to address this important issue. According to the reviewer's suggestion, we tried to knock down *c-Jun* in Th17(β) and Th17(23) cells. However, we are sorry that we could not get significant knockdown of *c-Jun* (data not shown), although we used 14 different shRNA or miRNA sequences against *c-Jun*, including the one used in a previous paper (*Immunity* 34, 741-754, 2011). Therefore, we analyzed effects of an inhibitor of JNK (c-Jun N-terminal kinase), which is an important regulator of c-Jun activity, on Th17(β) and Th17(23) differentiation (Supplementary Fig. 9). Treatment of the JNK inhibitor (SP600125) did not significantly affect *Rorc* induction on day 3 post activation in both *JunB*-deficient and control cells under Th17(β) conditions (Supplementary Fig. 9).

However, without data of *c-Jun*-knockout cells, we are not prepared to draw a firm conclusion relative to the relevance of *c-Jun* in pathogenic or non-pathogenic Th17 differentiation. We are preparing *c-Jun* conditional knockout mice to verify the significance of *c-Jun* in JunB-independent Th17 differentiation.

We have also tried to identify pathways responsible for TGF- β -dependent JunB-independent Th17 differentiation using inhibitors for TGF- β -signaling molecules; TGF- β receptor kinase inhibitor (SB43152), MEK inhibitor (PD98059), p38 inhibitor (SB203580), PI3 kinase inhibitor (LY294002), ROCK inhibitor (Y27632) or SMAD3 inhibitor (SIS3). The data showed that SB43152 treatment significantly inhibited *Rorc* induction in Th17(β) cells, regardless of the presence of JunB (Supplementary Fig. 9), indicating that TGF- β receptor kinase activity is essential for JunB-independent Th17 differentiation. However, there was only partial or no reduction of *Rorc* expression in Th17(β) cells treated with the other inhibitors. Thus, although TGF- β receptor kinase activity is important for JunB-independent Th17 differentiation, the downstream signaling pathways remain unknown.

Although we agree that identification of another signaling/transcription factor that mediates JunB-independent Th17 differentiation is important, we would like to reserve this issue for a future study, because we want to focus on functions of JunB in pathogenic Th17 generation in this manuscript.

Supplementary Fig. 9 was added in the current manuscript (page 13, lines 12-20). The method was also changed (page 15, lines 13-18).

- An issue is whether JUNB ablation locks expansion of any Th17 cells in vivo, not just pathogenic ones. The results obtained in the periphery during EAE and colitis suggest this possibility. Can the authors test expansion of non-pathogenic Th17 cells in a situation in which the described pathogenic Th17 cells are less important, perhaps a bacterial infection - or e.g. the response of gut Th17 cells to SFB, the Th17 cell inducing bacteria detected by the Littmann group?*

Our new data shows a substantial increase in proportions of IL-17A-producing CD4⁺ T cells in spleens of *Cd4^{Cre}JunB^{fl/fl}* mice until day 7 after immunization with MOG peptides (Supplemental Fig. 7d). Furthermore, we have investigated a role for JunB in an anti-CD3 antibody treatment model, in which Th17 cells are induced and migrate into the gut. Although we observed a substantial increase in proportions of CD4 T cells expressing IL-17A or ROR γ t in the LP of duodenum in *JunB*-deficient mice after anti-CD3 antibody treatment, the abundance was much lower than in control mice (Supplementary Fig. 8c). These results suggest that some Th17 cells can be generated independently of JunB even in inflammatory settings, although JunB is required for full development

of inflammatory Th17 cells.

On the other hand, a substantial number of Th17 cells are found in the guts of *JunB*-deficient mice at steady state (Fig. 5). Littman *et al.* have clearly shown that generation of this type of Th17 population depends on segmented filamentous bacteria (SFB). Thus, in a non-inflammatory environment, SFB-dependent Th17 cells seem to be generated in a JUNB-independent manner.

The following data were added in the current manuscript; Supplementary Fig. 7d (page 10, lines 11-13), Supplementary Fig. 8c (page 11, lines 7-13). Anti-CD3 antibody treatment procedure was also added (page 18, 25-27).

3. *I miss further analysis of the Th17 cells which are induced by IL-6 plus TGF β 3 which were also shown previously to be pathogenic, even independently of T-bet (Lee et al, NI, 2012). How is their transcriptional regulation compared to the Th17 (23) cells?*

We analyzed mRNA expression of *Il17a*, *Il17f*, *Rorc*, *Il23r*, *Foxp3*, and *Tbx21* in *JunB*-deficient cells activated in the presence of IL-6 and TGF- β 3. Loss of JunB resulted in a significant decrease in *Il17a* expression and an increase in *Foxp3* and *Tbx21*, while it had only minor effects, if any, on expression of *Il17f*, *Rorc*, and *Il23r* under Th17(β 3) conditions (Supplementary Fig 5 c). Thus, effects of JunB deficiency on differentiation of Th17(β 3) cells is similar to Th17(β 1) cells, but not Th17(23) cells.

The data were described in the current manuscript (page 7, lines 17-20).

4. *In Fig. 2b, the staining for ROR γ T is not convincing. Can this finding be repeated by a Western Blot?*

We optimized the staining protocol for ROR γ T. The new data show a significant increase of ROR γ T expression in control *JunB*^{fl/fl} cells under Th17(23) conditions compared to Th0 conditions (Fig.2b and Supplementary Fig. 10). However, there is no detectable induction of ROR γ T in *Cd4*^{Cre} *JunB*^{fl/fl} cells, even under Th17(23)-polarizing conditions (Fig.2b)

We also detected a severe reduction of ROR γ T expression in *JunB*-deficient Th17(23) cells by immunoblot analysis (Supplementary Fig. 4b).

Figure 2b was replaced, and Supplementary Figs. 4b (page 6, line 6) and 10 (page 15, lines 19-20) were added in the current manuscript.

5. *In sFig. 1b: Was this result obtained using Th17 conditions?*

We are sorry that we did not explain the conditions. We have now modified the sentence in the figure legend as follow (page 25, lines 11-14). Naive CD4⁺ T cells were activated under Th0 condition for 36 hr, infected with retrovirus expressing JunB or control shRNA, and then incubated for another 3 days in the presence of IL-6, IL-1 β , and IL-23.

Reviewer #2

- 1. This model is of great potential interest if authors will propose which factor(s) drive the expression of ROR γ t in non-pathogenic Th17 cells (instead of the IL-6/JunB axis), the cell type abundantly found in the gut of JunB deficient mice and in the in vitro cultures. The intriguing fact is that Th17 cell development in non-pathogenic conditions is also driven by the same set of transcription factors (namely IRF4, BATF and STAT3), activity of which is proposed here to be JUNB-dependent. Could it be IL-21? Or different sources of IL-6 are important?*

As we responded to comment #1 of reviewer #1, we have not yet been able to address this important issue. According to the reviewer's suggestion, we have investigated roles for IL-21 in JunB-independent Th17 differentiation. Addition of exogenous IL-21 to JunB-deficient cells activated under Th17(23) conditions did not rescue impairment of IL-17A induction (Fig. A). Furthermore, anti-IL-21 antibody treatment did not significantly affect IL-17A expression in JunB-deficient cells under Th17(b) conditions. Thus, IL-21 seems to be unnecessary for JunB-independent Th17 differentiation (Fig. A). We did not add these data in the current manuscript because IL-21 is not likely important for JunB-independent Th17 generation.

As mentioned above, we have shown that TGF- β receptor kinase activity is critical for JunB-independent Th17 differentiation (Supplementary Fig. 9), but the downstream signaling events remain to be determined. We agree that it is important to identify factors responsible for JunB-independent, non-pathogenic Th17 differentiation to achieve comprehensive understanding of Th17 differentiation. Therefore, we would like to address this issue using cells/mice deficient in TGF- β signaling molecules, such as c-Jun and SMAD2/3, in a future study.

Supplementary Fig. 9 was added in the current manuscript (page 13, lines 12-20).

Fig. A. Naïve CD4 T cells were activated under Th_H17(β) conditions with anti-IL-21 antibody (a) or Th_H17(23) conditions with IL-21 (b) for 3 days. IL-17A and IFN-γ expression was analyzed by FACS.

2. A main problem I see is that all conditions tested here included IL-6, which ultimately stimulated the expression of JunB and RORγt (not everywhere shown), leading to the generation of Th_H17 cells. However, effects of additionally added cytokines to mimic pathogenic (+ IL-1β + IL-23) and non-pathogenic (+ TGFβ1) conditions could be independent of JunB and do not allow to put IL-23 as a key signaling partner of JunB without additional experimental support. What would happen if IL-23 is added into non-pathogenic (IL-6/TGFβ1) conditions? Will it mimic effects found in IL-6/IL-1β/IL-23 conditions?

Apart of aforementioned concern and several minor points, which are listed below; the study is well structured and written. Overall the results are solid, convincing and shed significant light in our understanding of Th_H17-cell mediated pathology.

We greatly appreciate this important suggestion. Our immunoblot analysis showed that IL-6 stimulation was enough at least to facilitate expression of RORγt (Fig. B), Therefore, as the reviewer points out, the importance of JunB in IL-23-signaling was not conclusive in our previous manuscript. As IL-23 promotes pathogenicity of Th_H17(β) cells, we investigated a role for JunB in IL-23-stimulated Th_H17(β) cells. When we activated *JunB*-deficient CD4⁺ T cells in the presence of TGFβ1 and IL-6, with or without IL-23, IL-23 did not affect IL-17A or IFN-γ production in either *JunB*-deficient or control cells (Supplementary Fig. 4e).

We next examined impact of JunB deficiency on IL-23-stimulated T_H17(β) cells in the absence of TGF-β1. We activated *JunB*-deficient T cells under Th17(β) conditions for 3 days and sorted IL-17-high and IL-17-low populations using an IL-17-capture method for further culturing with IL-23 alone. Although JunB deficiency did not affect expression of IL-17A upon re-stimulation immediately after sorting, further culturing of IL-17A-high cells with IL-23 significantly decreased the abundance of IL-17A-expressing cells in *JunB*-deficient cells, but not in controls (Fig. 2e and supplementary Fig. 4f). The data also showed a marked increase in a proportion of IL-17A⁺IFN-γ⁺ cells in the absence of JunB (Fig. 2e). Interestingly, IL-23 facilitated IL-17A expression in control IL-17A-low cells, but not in *JunB*-deficient cells (Fig. 2e). Thus, JunB seems to play an important role in IL-23-dependent differentiation and maintenance of Th17 cells in the absence of TGF-β1.

The following data were added in the current manuscript; Fig. 2e and Supplementary Fig. 4f (page 6, lines 21-30 and page 7, lines 1-3) and Supplementary Fig. 4e (page 6, lines 18-21) were added in the current manuscript. The IL-17-capture method was added (page 16, lines 1-6). We did not include Figure B in the revised manuscript because we have shown the role for JunB in the induction of RORγt in Th17(23) cells, but not in cells stimulated with IL-6 alone, in this study.

Fig. B. Naïve CD4 T cells were activated in the presence of IL-6, IL6+IL-1β+IL-23 (Th17(23)), or IL-6+TGF-β (Th17 (β)) for 3 days, and RORγt expression was detected by immunoblot analysis.

3. *Suppl Fig 1. Authors used Enpp2 mRNA levels as an indicator of Th17 cells development. Not clear why they choose this marker over broadly accepted Th17-defining products such as Rorc, IL-17 etc. ? The fact that knockdown of JunB on Th17 cells resulted in reduction of Enpp2 levels 3 days after transfection speaks more in favor of the maintenance requirements, rather than induction of Th17 cell development as they claim (Suppl Fig 1b).*

We used *Enpp2* expression as a readout of the screening because *Enpp2* induction was

dependent on IL-23 stimulation. However, we also evaluated the effect of JunB knockdown on induction of *Ii23r*, a Th17 signature gene, and found that knockdown of JunB resulted in a significant reduction of *Ii23r* expression (Supplementary Fig. 1c).

In our RNAi assay, naive CD4⁺ T cells were activated under Th0 conditions for 36 hr and infected with shRNA-expressing retroviruses, followed by culturing under Th17(23) conditions. Therefore, as the reviewer suggests, it is possible that JUNB may be required not only for induction, but also for maintenance of Th17(23) cells. However, because we have not analyzed timing of JunB knockdown, we do not wish to mention that possibility in this manuscript.

Supplementary Fig. 1c was added in the current manuscript (page 4, lines 29-30).

4. *Fig 1. Since IL-6 alone is enough to induce strong JunB expression and neither IL-1 nor IL-23 increased this effect, how IL-23 promotes pathogenic Th17 cell development in an JunB-dependent manner as authors claim? Authors need to provide more evidences to distinguish IL-6/IL-23 discrepancy.*

As we mentioned above, we have now provided new data showing that JUNB is important for IL-23-dependent Th17 differentiation/maintenance (Fig. 2e and Supplementary Fig. 4f; page 6, lines 21-30 and page 7, lines 1-3).

5. *Fig 2a-d. Data presented here show that IL-6 fail to suppress Tregs and Th1 development in JunB deficient background. Does JunB regulate IL-6R expression or it serves as direct suppressor of Foxp3 and T-bet/Eomes? What is the relevance of discovered strong ROR γ t-Foxp3 co-expression without IL-17 production here and in in vivo system?*

Our qRT-PCR data showed that loss of JunB did not affect *Il6r* expression (Fig. C). Our ChIP-Seq data did not show significant DNA-binding of JunB as well as BATF, IRF4 at the *Foxp3* locus (Fig. D), suggesting JunB may regulate Foxp3 expression indirectly. On the other hand, JunB co-localized with BATF and IRF4 at the *Tbx21* and *Eomes* loci, suggesting that JunB may directly regulate these genes in Th17 cells (Fig. D). However, because we have not investigated detail mechanisms of JunB-dependent regulation of *Foxp3*, *Tbx21* and *Eomes* expression, we did not include Figure D in the revised manuscript.

Foxp3 suppresses ROR γ t-dependent IL-17A production in T_H17 cells (Nature, 2008, 453, 236-248). Indeed, the Foxp3-low population expressed higher levels of IL-17A than the Foxp3-high population in *JunB*-deficient cells under Th17(β) conditions (Supplementary Fig. 4d). This suggests that a partial defect in IL-17A production

in *JunB*-deficient $T_{H17}(\beta)$ cells, which we observed in Fig. 2a, might be due to aberrant induction of *Foxp3*.

Although we cannot fully explain the reason of discrepancy between *in vitro* and *in vivo* gut data regarding cells co-expressing $ROR\gamma t$ and *Foxp3*, we speculate that *Foxp3* expression might be unstable compared to $ROR\gamma t$ in Th17 cells generated in a *JunB*-independent manner.

Supplementary Fig. 4d was added in the current manuscript (page 6, lines 13-17). As IL-6 receptor expression is not regulated by *JunB*, we did not include the Figure C in the manuscript. We also did not include Figure D in the manuscript because we have not investigated detail mechanisms of *JunB*-dependent regulation of *Foxp3*, *Tbx21* and *Eomes* expression.

Fig. C. Naïve CD4 T cells were activated under Th17(23) conditions for 3 days. *Ii6r* mRNA expression was measured by qRT-qPCR.

Fig. D. ChIP-Sequencing data for loci of *Tbx21*, *Eomes*, and *Foxp3*.

6. *FACS plots should be supplemented with total cell numbers in order to support the claim of developmental defects and not increased plasticity of JunB deficient Th17 cells towards other lineages. Changing of X- and Y-axis on FACS plots (panel b) will make easier the comparison between IL-17A and ROR γ t expression (the same is for the panel d).*

We have now provided data regarding total cell numbers for the *in vivo* FACS analysis (Supplementary Fig. 7a, c). This shows that a substantial number of gut-resident Th17 cells are generated in the absence of JunB. In terms of Th17 plasticity, we have now provided Th17 fate-mapping data (Fig. 6d and Supplementary Fig. 7e). Details are discussed below (Response to the comment#10). We have also changed the X and Y axis in panel B (Fig. 2), as the reviewer suggested.

The following data were added in the current manuscript; Fig. 6d and Supplementary Fig. 7e (page 10, lines 15-27), Supplementary Figs 7a (page 9, 17-20) and Supplementary Fig. 7c (page 10, lines 7-11).

7. Fig 2e. Authors claim that *JunB* deficient Th17 cells generated in non-pathogenic conditions lose their IL-17 expression after re-stimulation with IL-23. This statement requires additional clarification. What happened to the initial Th17(β) cells, they died or changed their cytokine profile? In other words, do abundant IFN γ expression found in *JunB* deficient cultures supplemented with IL-23 originated from exTh17 cells? Analysis of cell numbers may help in answering this question if authors do not have Th17-fate mapping system in present.

As mentioned above, we sorted IL-17-high and IL-17A-low populations generated under Th17(β) conditions by an IL-17-capture method. The data clearly showed that a subset of initial IL-17A-expressing Th17(β) stopped expressing IL-17A and produced IFN- γ (Fig. 2e). We also did not observe increased cell death in *JunB*-deficient cells (Supplementary Fig. 4f).

The following data were added in the current manuscript; Fig. 2e and Supplementary Fig. 4f (page 6, lines 21-30 and page 7, lines 1-3)

Fig 3. Does IL-10 expression dependent on JunB? Please discuss this point.

Expression of *Il10*, which is regulated by BATF and IRF4 (Nature 490, 543-546, 2012), was induced in a *JunB*-independent manner (Supplementary Fig. 5b). At the *Il10* locus, there was enrichment of BATF, IRF4, and *JunB* in both Th17(β) and Th17(23) cells (Supplementary Fig. 6b), which suggests that *JunB* colocalizes with BATF and IRF4 at a large number of gene loci, but that a limited subset of the genes is regulated by *JunB*.

The following data were added in the current manuscript; Supplementary Figs. 5b (page 8, lines 17-20) and 6b (page 8, 20-23)

8. Fig 4d. Authors state: “Loss of *JUNB* considerably diminished DNA-binding of BATF, IRF4, and STAT3 at the *Rorc* locus, under TH17(23) conditions, but not under TH17(β) conditions (Fig. 4d).”

The corresponding figure shows no changes under Th17(β) conditions only for BATF, while IRF4 and STAT3 input indicated to be significantly decreased. Explain this discrepancy.

We have now explained the differential effects on BATF and IRF4/STAT3 in the manuscript (Fig. 4d). The data suggest that *JUNB*/BATF might interact with IRF4 and STAT3 more readily than BATF dimers with other AP-1 family members, although the biological meaning of this interaction remains unclear.

We have corrected the sentence in the manuscript (page 9, lines 2-6).

9. *Fig 5. As a suggestion, authors may want to use an anti-CD3 treatment model, which is proposed to be strongly dependent on IL-6/TGF β to strengthen non-pathogenic Th17 cells development in in vivo system.*

As mentioned in response to Reviewer #1, we have added data about the anti-CD3 antibody treatment model. JUNB deficiency greatly reduced numbers of duodenum-infiltrating Th17 cells. Nevertheless, we could see a substantial increase in cells expressing IL-17A or ROR γ t after anti-CD3 antibody treatment in the gut of *JunB*-deficient mice (Supplementary Fig.8c). This suggests that at least a subset of Th17 cells can be generated in a JUNB-independent manner in this model.

Supplementary Fig. 8c was added in the current manuscript (page 11, lines 7-13).

10. *Fig 6. Authors need to show numbers of CNS infiltrating CD4 T cells and especially Th1 cells (shown in panel c), to support developmental failure over the migration disadvantage or aberrant plasticity.*

We have now shown the numbers of CNS-infiltrating CD4 T cells (Supplementary Fig.7c). There are significantly fewer Th1 cells in *JunB*-deficient mice than controls (Supplementary Fig7c). We speculate that this reduction may be due to defects in generation of pathogenic Th17 cells, which might recruit or trans-differentiate into Th17 cells in the CNS.

In terms of plasticity of Th17 cells, we have now provided data of Th17 fate-mapping experiments and shown that loss of JunB reduced IL-17A/IFN- γ double-producing cells derived from Th17 cells, but had almost no effect on IL-17A single-producing cells (Fig. 6d and Supplementary Fig. 7e), suggesting that defects in generation of Th17 cells observed in *JunB*-deficient mice might not be due to increased plasticity of Th17 cells. Furthermore, in an inflammatory context, JunB is likely required for generation of Th17 cells that are competent to differentiate to IL-17A/IFN- γ double-producing cells (Fig. 6d). This is similar to reported phenotype of *Ii23p19*-deficient Th17 fate-mapping reporter mice (Nature Immunology 12, 255-263, 2011). Taken together, these results further support our model that JUNB is critical for IL-23-dependent pathogenic Th17 generation.

The following data were added in the current manuscript; Supplementary Fig. 7c (page 10, lines 7-11), Fig. 6d and Supplementary Fig. 7e (page 10, lines 15-27).

11. *In many in vivo experiments the gating strategy is not clear. Did authors use any*

markers to define T cells? Gating on only CD4 positive cells seems to be inappropriate, especially for the gut samples normally enriched with CD4+ ILC's.

We have now provided all flow cytometry gating strategies in Supplementary Fig. 10 (page 15, lines 19-20). We used anti-CD3 antibody to distinguish T cells from ILCs (Supplementary Fig. 10).

12. Authors should state the *JunB* expression levels in a novel *JunB* conditional KO mice, used as a controls, in comparison to C57BL/6 wild type mice.

We have shown this in the new supplementary Fig2b. Our floxed *JunB* mice express comparable levels of JunB compared to C57BL/6 wild type mice.

13. Standardize the gene symbols used in the methods part.

Done. The following have been standardized; *JunB* gene, JunB protein, *JunB*^{fl/fl}, *Cd4*^{Cre}, IFN- γ protein, *Stat3* gene, GFP protein,

14. In the introduction: "Furthermore, TGF- β 1 diminishes pathogenicity of TH17 cells isolated from EAE-induced mice, while IL-23 makes TH17(β) cells pathogenic, suggesting that TGF- β 1 and IL-23 signals facilitate differentiation of non-pathogenic and pathogenic TH17 cells, respectively."

Please re-write. It is broadly accepted that IL-23 is needed for Th17 cell pathogenicity and can convert non-pathogenic Th17 cells into pathogenic.

We have modified the sentence as the reviewer suggested (page 3, lines 13-15).

15. In the discussion: "ROR γ t regulates induction of a small subset of TH17 genes." Word restricted or certain seems to be more appropriate in this context.

We have corrected the sentence according to the reviewer's suggestion: a "restricted" subset (page 12, line21).

Reviewer #3

1. This manuscript by Hasan et al revisits a currently much debated issue focused on transcriptional control of pathogenic vs non-pathogenic Th17 cells. There has

been a flurry of papers in the last couple of years attempting to identify markers of such functional states in Th17 cells with a view of pinpointing potential therapeutic targets in pathogenic Th17 responses.

As such the present manuscript is not particularly original, but adds another facet to the ongoing debate. Similar to previous approaches the authors focus on in vitro generated Th17 cells using two different protocols that are reported to generate either pathogenic or non-pathogenic Th17 cells. There are numerous permutations for in vitro generation of Th17 cells with many associated deficiencies and none of them fully reflect the Th17 effector state in vivo. A particular problem for these approaches is the heterogeneous nature of the cells growing out of these cultures and in the absence of a reporter system it is simply not clear how many of them are actually Th17 cells.

We appreciate the reviewer's useful comments. In this revised manuscript, we have performed 1) microarray analysis using purified IL-17-high and IL-17-low-populations in *JunB*-deficient cells, 2) analysis of plasticity/stability of Th17 cells using a Th17 fate-mapping reporter system. The new data further support our main claim that JunB is important for generation of pathogenic Th17 cells. We have discussed this below.

- 2. It is possible though that the effects of JUNB are related to plasticity vs stability rather than pathogenicity or non-pathogenicity. It would be preferable to use a reporter or better even fate reporter to address these issues and avoid the in vitro protocols which are not without controversy.*

According to the reviewer's suggestion, to explore a role for JunB in Th17 plasticity and stability, we performed Th17 fate-mapping analysis. We induced EAE in *JunB*-deficient Th17 fate-mapping reporter (*Il17a^{cre}R26R^{eYFP}JunB^{fl/fl}*) mice, in which constitutive eYFP expression and *JunB*-deficiency are induced in cells expressing IL-17A. Our data showed that a proportion of eYFP⁺CD4⁺ T cells in *JunB*-deficient reporter mice was slightly lower than controls (*Il17a^{cre}R26R^{eYFP}JunB^{fl/+}*), although not statistically significant, on day 7 after immunization with MOG35-55 peptides (Supplementary Fig. 7d). Furthermore, in the eYFP⁺CD4⁺ T cell population, *JunB* deficiency resulted in a significant reduction in a proportion of IL-17A/IFN- γ double-producing cells, but had almost no effect on IL-17A single-producing cells (Fig. 6d and Supplementary Fig. 7e), suggesting that defects in generation of Th17 cells observed in *JunB*-deficient mice might not be due to increased plasticity of Th17 cells.

Furthermore, in an inflammatory context, JunB is likely required for generation of Th17 cells which are competent to differentiate into IL-17A/IFN- γ double-producing cells. Given that a similar phenotype was observed in *Il23p19*-deficient Th17-fate mapping reporter mice (Nature Immunology, 2011, 12, 255-263), JunB might play a role in IL-23-dependent pathogenic Th17 generation *in vivo*. This also suggests that pathogenicity and stability/plasticity of Th17 cells may not be mutually exclusive. Rather, pathogenic Th17 cells are likely stable to expand the population size in an inflammatory setting, and their plasticity might promote deviation toward a Th1-like phenotype at a certain phase of Th17 differentiation, which probably contributes to induce a highly inflammatory environment.

The following data were added in the current manuscript; Supplementary Fig. 7d (page 10, lines 11-13), Fig. 6d and Supplementary Fig. 7e (page 10, lines 15-27).

3. *It is quite evident from Fig. 2 that absence of JUNB allows the emergence of Tregs and Th1 in vitro. So does JUNB regulate ROR γ t induction or rather restrain Treg/Th1 differentiation?*

To examine whether significant impairment of ROR γ t induction is due to aberrant induction of T-bet in JunB-deficient cells under Th17(23) conditions, we analyzed ROR γ t and T-bet expression at the single cell levels by flow cytometry. The data showed that ROR γ t expression was significantly reduced not only in the T-bet-high population but also in the T-bet-low population in *JunB*-deficient cells under Th17(23) conditions (Supplementary Fig. 5f), suggesting that JunB regulates expression of ROR γ t and T-bet independently.

JunB deficiency also results in a marked induction of Foxp3, but does not affect ROR γ t expression, under Th17(β) conditions (Fig. 2b). Our new data have shown that Foxp3 expression in IL-17-low population is significantly higher than IL-17-high population in *JunB*-deficient cells under Th17(β) conditions (Supplementary Figs. 4d and 5e), suggesting that Foxp3 might inhibit IL-17A expression. On the other hand, ROR γ t induction is diminished without aberrant induction of Foxp3 in the absence of JunB under Th17(23) conditions (Fig. 2b). These data suggest that JunB regulates ROR γ t expression independently of inhibition of Foxp3 expression, and that Foxp3 regulates expression of IL-17A, but not ROR γ t.

On the other hand, our results clearly show that JunB is required for ROR γ t induction (Fig. 2b) and DNA-binding of BATF, IRF4, and STAT3 at the *Rorc* locus under Th17(23) conditions (Fig. 4d), implying a direct regulation of ROR γ t induction by JunB.

The following data were added in the current manuscript; Supplementary Fig. 5f (page 7 lines 28-31 and page 8, lines 1-3), Supplementary Fig. 4d (page 6, lines 13-17), Supplementary Fig. 5e (page 7, lines 25-27).

4. *ROR γ t staining in Fig. 2b (Th17(23) looks essentially negative also in the control group (JUNBf1/f1).*

As we responded to reviewer #1, we have now provided new FACS data showing a significant induction of ROR γ t under Th17(23) conditions. We also confirmed this by western blot analysis in Supplementary Fig. 4b (page 6, line 6).

5. *It is difficult to interpret the microarray results, which are based on in vitro cultures of very heterogeneous populations between the two conditions.*

We agree that heterogeneity of Th17 cells generated *in vitro* makes it difficult to interpret the data. Therefore, we investigated effects of JunB deficiency in both IL-17-high and IL-17-low populations induced under Th17(β) conditions. We found that JunB deficiency decreased expression of a subset of common T_H17 genes, such as *Ccr5* in both IL-17-high and IL-17-low populations in a similar manner (Supplementary Fig. 5d, e). However, expression of *Foxp3*, but not *Rorc*, was significantly higher in IL-17-low population than IL-17-high population (Supplementary Fig. 5e), suggesting that Foxp3 may suppress IL-17 expression at the transcriptional level. Thus, JunB seems to be involved not only in ROR γ t-dependent Th17 transcription, but also in regulation of a subset of genes independently of ROR γ t

Supplementary Fig. 5d and e were added in the current manuscript (page 7, lines 21-27).

6. *The in vivo data on gut resident Th17 cells are interesting, suggesting that JUNB has no impact on their generation. However, the fact that Foxp3+ cells were reduced in the absence of JUNB is in direct contradiction to the in vitro results, suggesting that it is problematic to extrapolate from in vitro cultures of Th17 cells to in the in vivo scenario.*

As the reviewer points out, our data show a significant reduction of Foxp3+ T cells in the small intestinal lamina propria (SI LP). We speculate that this might be due to a role for JunB in Treg differentiation. In the SI LP, there are heterogeneous Treg sub-populations, including peripherally derived Tregs and thymus-derived Tregs (Nature Reviews Immunology 16, 295-309, 2016). As JunB plays a key role in generation of a subset of Th17 cells, we are interested in whether JunB is also involved in differentiation of a subset of Tregs. However, we want to leave this interesting question for a future study.

We did not mention this in the current manuscript.

7. *With respect to the EAE data in Fig. 6 it is evident that a Th17 response is simply never initiated- not a scenario of non-pathogenic vs pathogenic induction.*

We have now shown that CD4⁺IL-17⁺ T cells slightly increased in spleens of both *Cd4^{Cre}JunB^{fl/fl}* mice and controls until day 7 after EAE induction, and then decreased in *Cd4^{Cre}JunB^{fl/fl}* mice, but not in controls (Supplementary Fig. 7d). We also used an anti-CD3 antibody treatment model and showed that injection of anti-CD3 antibody increased CD4⁺ T cells expressing IL-17A and ROR γ t in the LP of duodenum in both *Cd4^{Cre}JunB^{fl/fl}* and control mice, although the abundance of these cells was much lower in *Cd4^{Cre}JunB^{fl/fl}* mice than in control mice (Supplementary Fig. 8c). Thus, JunB-independent Th17 response seems to be at least partly initiated even in inflammatory settings.

The following data were added in the current manuscript; Supplementary Fig. 7d (page 10, lines 11-13), Supplementary Fig. 8c (page 11, lines 7-13).

8. *Given that JUNB deficiency seems to strongly affect expression of IL-23R (Fig. 3) this is not surprising as EAE is not induced in the absence of IL-23 signals, shown initially by Dan Cua' s work. Without a Th17 response that exceeds a certain threshold there is no infiltration of other cells types in the CNS, but the sequelae of this have been described before and there are many genetic modifications of the Th17 program that result in the same phenotype= lack of CNS pathology in this model.*

JunB deficiency significantly impairs *Il23r* expression under Th17(23) conditions, but little under Th17(β) conditions (Fig 3c). Furthermore, JUNB seems to be important for IL-23-dependent differentiation and maintenance of TGF- β /IL-6-induced Th17 cells, which express relatively normal levels of IL-23R, in the absence of TGF- β 1 (Fig. 2e). Based on these *in vitro* data, we agree that *JunB*-deficient mice are resistant to EAE probably because of defects in IL-23 signaling. We also agree that, similar to other critical regulators for Th17 differentiation including BATF, IRF4 and ROR γ t, *JunB* deficiency results in a severe impairment of generation of CNS-infiltrating pathogenic Th17 cells, which might cause recruitment of other immune cells into the CNS.

We did not mention this in the current manuscript.

9. *So altogether this manuscript identified another important player in the Th17 pathway, but the attempts to link this to decisions regarding pathogenicity or not are not convincing and there is a lack of connection between observations in vitro and in vivo data that further complicate an already difficult issue.*

As discussed above, Th17 fate-mapping data has suggested that defects in generation of Th17 cells observed in JunB-deficient mice might not be due to increased plasticity of Th17 cells and that JunB is likely required for generation of Th17 cells which are competent to differentiate to IL-17A/IFN- γ double-producing cells.

We agree that there is still a big gap between Th17 cells generated *in vitro* and *in vivo*. In particular, the time frames are quite different. We have focused on early Th17 differentiation (within 4 days) *in vitro*, but Th17 cells, which we observed *in vivo*, probably go through additional events (encountering many other environmental factors, including cytokines) during full differentiation, which takes more than 10 days in the case of EAE. Nevertheless, some cytokines (IL-6, TGF- β 1, IL-1 β , and IL-23) and transcription factors (BATF, IRF4, STAT3, and ROR γ t), which are important for Th17 differentiation *in vitro*, are crucial for *in vivo* Th17 generation, suggesting that *in vitro* Th17 differentiation may recapitulate certain priming events of *in vivo* Th17 differentiation. Therefore, we speculate that JunB-dependent generation of Th17 cells in the absence of TGF- β , which we observed *in vitro*, might be relevant to *in vivo* pathogenic Th17 differentiation. We believe that the selective requirement of JunB for differentiation of a pathogenic Th17 subset offers new insight into the complicated transcriptional mechanisms regulating differentiation of heterogeneous Th17 populations.

We did not mention this in the current manuscript.

Responses to reviewers' comments.

We appreciate the reviewers' helpful comments to improve our manuscript. Below please find our point-by-point responses to questions and comments. All changes in the manuscript are marked in red.

Reviewer #1 (Remarks to the Author):

I think that the authors did a good job to address the raised issues. They could not clarify everything, but the topic is complicated indeed - and the new information is significant. I have only some small points left:

Line 720, 793: SD of what exactly? Of the two experiments?

We have added the information (lines 743 and 816).

Fig. 1c,d: also naive CD4⁺ T cells?

We have now added Supplementary Fig. 1a showing JunB expression in naive CD4 T cells and have explained it (lines 143-144, 816-818).

In suppl. Fig. 4, it is striking that JunB expression is much less in Th17 (23) than in Th17 (β) cells. This is in striking contrast to Fig. 1. Why? How often was this performed?

Expression of ROR γ t, but not JunB was shown in the previous Supplementary Fig. 4b. JunB expression levels are comparable between Th17(β) and Th17(23) cells (Fig. 1a,b). Western blot data of ROR γ t (New main figure 2c) are representative of two independent experiments.

L 169, „not“ does not reflect the data. „Less striking“ would be better.

We have modified the sentence as the reviewer suggested (line 170).

The legend for Fig. 3a is still quite hard to understand for me. It uses the values of suppl 5a but compares ko vs wt cells, correct? This should be clarified.

We have now clarified that the data show fold changes in ko vs wt and that Fig. 3a and supplementary Fig. 5a are based on the same data (lines 765-766 and 847-850).

L 323 these are not Cd4CreJunBfl/fl mice, but RAG ko mice having been transplanted with Cd4CreJunBfl/fl cells??

Yes. We have corrected the sentence (lines 325-326).

Reviewer #2 (Remarks to the Author):

The authors need to familiarize themselves with the developmental/maintenance issue of JunB deficient cells. I agree that presented data is not firmly enough to claim its necessity for Th17 cells maintenance/plasticity, however it is a very likely additive scenario and authors need to mention it at least in the discussion part.

We agree with the comment. According to the reviewer's suggestion, we have discussed the role of JunB in maintenance and plasticity of TH17 cells (242-246, 375-391).

Moreover, in light of a new data, ALL effects of JunB deficiency seem to be a result of impaired IL-23R expression, and this needs to be clearly stated in the title and abstract. In brief, IL-6 facilitates IL-23R expression in a JunB dependent manner. To note, IL-23 is not a differentiation factor for Th17 cells, unless authors wish to challenge this paradigm.

We agree that impaired IL-23R may cause the reduction of ROR γ t/IL-17 expression. Therefore, we have mentioned this possibility in the Discussion (lines 396-401). We also stated the necessity of IL-6-induced JunB for IL-23R expression in TH17 cells in the Abstract (lines 46-48). In addition, considering IL-23 functions in TH17 cells, we have changed the title (lines 4-5) and used "pathogenicity of TH17 cells" instead of "pathogenic TH17 differentiation" (lines 53, 87, 109-110, 112-113, 118, 295, 337-338, 344, and 434).

Minor points:

1) Make clear in the text what is the Enpp2 and why it was used as a readout for Th17 cells development.

We have added the explanation (lines 128-129).

2) Supplementary Fig. 4b. From the letter: "Indeed, the Foxp3-low population expressed higher levels of IL-17A than the Foxp3-high population in JunB-deficient cells under Th17(β) conditions (Supplementary Fig. 4d)".

Not convincing. Use MFI for IL-17A staining to claim the difference. The reference to this figure in the manuscript is not appropriate.

We have changed the sentence and added a supplemental figure showing the MFI (lines 182-188). We have also added another reference and modified the sentence to mention that Foxp3 inhibits IL-17A expression, which we think is related to Fig. 4d (lines 182-183 and 703-705).

3) Figure C supports unaffected initial IL-6 response by JunB deficient cells and can be used as supplementary figure, after explanation of the statistical significance in Th17(23) conditions.

We have shown the result in supplementary figure 4c and explained it (lines 171-175).

4) Fig 2b. Such dim ROR γ t staining cannot be shown in a main figure.

According to this suggestion, we have used ROR γ t western blot data for the main figure and moved the previous Figs. 2b and 2d to supplementary Figs. 4b and 4e, respectively (lines 753-756, 833-839).

5) Fig. 6d. Authors write in the manuscript “proportion of eYFP+CD4+ T cells in JunB-deficient reporter mice was slightly lower than controls”.

Four times decrease cannot be called a slight decrease, although could be not significant. Overall, day 7 after immunization is not the best time to investigate the plasticity by using fate-mapping system, later time points (and ideally CNS analysis) seems to be more appropriate for that purpose.

We agree with the comment and analyzed the cells on day 14 after EAE induction. We have used the new data in Figs. 6d and 6e (lines 311-314).

6) Supplementary Fig 7e. Did I miss the legend for this panel?

We are sorry that we did not have the legend. We have now added it (lines 877-879).

Reviewer #3 (Remarks to the Author):

The authors have made substantial efforts to answer the reviewer's queries.

In particular the inclusion of in vivo data with the IL-17 fate reporter answered an important question and points towards a function of JunB in IL-23 mediated pathogenicity. The remaining problem I see is the disconnection with the in vitro data. Here the authors claim that JunB deficiency leads to an increase in IFN γ and GM-CSF- in direct contradiction to what the fate reporter indicated in vivo.

Furthermore, the *in vivo* data show that the initial generation of Th17 cells that produce IL-17 is not suppressed, but the cells decline after the initial activation in line with what was found in IL-23 deficient mice.

However, *in vitro* JunB deficiency is claimed to compromise RORgt expression and thereby IL-17 with varying effects depending on the somewhat artificial culture conditions. The authors concede that there are ‘gaps’ between Th17 cells generated *in vitro* and *in vivo*, but chose not to discuss these issues in the manuscript. Using the fate reporter for the *in vitro* experiments would have established that the increase in IFN γ and GM-CSF comes most likely from alternative T cell fates as anti-CD3 activates any T cell depending on subsequent cytokine conditions and transcriptional programs. As the authors show RORgt and IL-17a defective under Th17 (23) polarizing conditions, which are equated in the literature with generation of ‘pathogenic’ Th17 cells, this does not link to the largely unperturbed Th17 cell induction *in vivo*, which wanes before the cells undergo IL-23 dependent plasticity towards IFN γ and/or GM-CSF secretion. There should at least be a critical discussion of these discrepancies in the manuscript.

We agree with the comment and have discussed the role of JunB in IL-23-dependent plasticity of TH17 cells and the discrepancy between *in vitro* and *in vivo* TH17 cells (lines 242-246, 375-391).